# Efficient Learning of Discrete Graphical Models

**Marc Vuffray, Sidhant Misra, Andrey Y. Lokhov**
Theoretical Division,
Los Alamos National Laboratory, USA
{vuffray, sidhant, lokhov}@lanl.gov

## Abstract

Graphical models are useful tools for describing structured high-dimensional probability distributions. Development of efficient algorithms for learning graphical models with least amount of data remains an active research topic. Reconstruction of graphical models that describe the statistics of *discrete* variables is a particularly challenging problem, for which the maximum likelihood approach is intractable. In this work, we provide the first sample-efficient method based on the *Interaction Screening* framework that allows one to provably learn fully general discrete factor models with node-specific discrete alphabets and multi-body interactions, specified in an arbitrary basis. We identify a single condition related to model parametrization that leads to rigorous guarantees on the recovery of model structure and parameters in any error norm, and is readily verifiable for a large class of models. Importantly, our bounds make explicit distinction between parameters that are proper to the model and priors used as an input to the algorithm. Finally, we show that the Interaction Screening framework includes all models previously considered in the literature as special cases, and for which our analysis shows a systematic improvement in sample complexity.

## 1 Introduction

Representing and understanding the structure of direct correlations between distinct random variables with graphical models is a fundamental task that is essential to scientific and engineering endeavors. It is the first step towards an understanding of interactions between interleaved constituents of elaborated systems [11]; it is key for developing causal theories [6]; and it is at the core of automated decision making [8], cybersecurity [5] and artificial intelligence [19].

The problem of reconstruction of graphical models from samples traces back to the seminal work of [7] for tree-structured graphical models, and as of today is still at the center of attention of the learning community. For factor models defined over general hypergraphs, the learning problem is particularly challenging in graphical models over discrete variables, for which the maximum likelihood estimator is in general computationally intractable. One of the earlier tractable algorithms that has been suggested to provably reconstruct the structure of a subset of pairwise binary graphical models is based on inferring the sparsity pattern of the so-called regularized pseudo-likelihood estimator, equivalent to regularized logistic regression in the binary case [16]. However, additional assumptions required for this algorithm to succeed severely limit the set of pairwise binary models that can be learned [15]. After it was proven that reconstruction of any discrete graphical models with bounded degree can be done in polynomial time in the system size [4], Bresler showed that it is possible to bring the computational complexity down to quasi-quadratic in the number of variables for Ising models (pairwise graphical models over binary variables); however, the resulting algorithm has non-optimal sample requirements that are double-exponential in other model parameters [3]. The first computationally efficient reconstruction algorithm for sparse pairwise binary graphical models with a near-optimal sample complexity with respect to the information theoretic lower bound [17],

called RISE, was designed and analyzed in [18]. The algorithm RISE suggested in this work is based on the minimization of a novel local convex loss function, called the *Interaction Screening* objective, supplemented with an $\ell_1$ penalty to promote sparsity. Even though it has been later shown in [13] that regularized pseudo-likelihood supplemented with a crucial post-processing step also leads to a structure estimator for pairwise binary models, strong numerical and theoretical evidence provided in that work demonstrated that RISE is superior in terms of worst-case sample complexity.

Algorithms for learning discrete graphical models beyond pairwise and binary alphabets have been proposed only recently in [9] and [12]. The method in [9] works for arbitrary models with bounded degrees, but being a generalization of Bresler's algorithm for Ising models [3], it suffers from similar prohibitive sample requirements growing double-exponentially in the strength of model parameters. The so-called SPARSITRON algorithm in [12] has the flavor of a stochastic first order method with multiplicative updates. It has a low computational complexity and is sample-efficient for structure recovery of two subclasses of discrete graphical models: multiwise graphical models over binary variables or pairwise models with general alphabets. A recent follow-up work [20] considered an $\ell_{2,1}$ constrained logistic regression, and showed that it provides a slight improvement of the sample complexity compared to [12] in the case of pairwise models over non-binary variables.

In this work, we propose a general framework for learning general discrete factor models expressed in an arbitrary parametric form. Our estimator termed GRISE is based on a significant generalization of the Interaction Screening method of [18, 13], previously introduced for pairwise binary models. Our primary insight lies in the identification of a single general condition related to model parameterization that is sufficient to obtain bounds on sample complexity. We show that this condition can be reduced to a set of local identifiability conditions that only depend on the size of the maximal clique of the factor graph and can be explicitly verified in an efficient way. We propose an iterative algorithm called SUPRISE which is based on GRISE and show that it can efficiently perform structure and parameter estimation for arbitrary graphical models. Existing results in the literature on this topic [18, 9, 12, 20] can be obtained as special cases of our general reconstruction results, which noticeably includes the challenging case of multi-body interactions defined over general discrete alphabets. Our theoretical guarantees can be expressed in any error norm, and explicitly includes distinction between bounds on the parameters of the underlying model and the prior parameters used in the optimization; as a result prior information that is not tight only has moderate effect on the sample complexity bounds. Finally, we also provide a fully parallelizable algorithmic formulation for the GRISE estimator and SUPRISE algorithm, and show that they have efficient run times of $\widetilde{O}(p^L)$ for a model of size $p$ with $L$-order interactions, that includes the best-known $\widetilde{O}(p^2)$ scaling for pairwise models.

## 2 Problem formulation

In this Section, we formulate the general discrete graphical model selection problem that we consider and describe conditions that makes this problem well-posed.

### 2.1 Parameterized family of models

We consider positive joint probability distributions over $p$ variables $\sigma_i \in \mathcal{A}_i$ for $i = 1, \ldots, p$. The set of variable indices $i$ is referred to as vertices $\mathcal{V} = 1, \ldots, p$. Node-dependent alphabets $\mathcal{A}_i$ are assumed to be discrete and of size bounded by $q > 0$. Without loss of generality, the positive probability distribution over the $p$-dimensional vector $\underline{\sigma}$ can be expressed as

$$\mu(\underline{\sigma}) = \frac{1}{Z} \exp\left(\sum_{k \in \mathcal{K}} \theta_k^* f_k(\underline{\sigma}_k)\right), \tag{1}$$

where $\{f_k, \ k \in \mathcal{K}\}$ is a set of *basis functions* acting upon subsets of variables $\underline{\sigma}_k \subseteq \underline{\sigma}$ that specify a family of distributions and $\theta_k^*$ are parameters that specify a model within this family. The quantity $Z$ denotes the partition function and serves as a normalization constant that enforces that the $\mu$ in (1) is a probability distribution. For $i \in \{1, \ldots, p\}$, let $\mathcal{K}_i \subseteq \mathcal{K}$ denote the set of factors corresponding to basis functions acting upon subsets $\underline{\sigma}_k$ that contain the variable $\sigma_i$ and $|\mathcal{K}_i| = \mathbf{K}_i$.

Given any set of basis functions, we can locally center them by first defining for a given $i \in [p]$, the *local centering functions*

$$\phi_{ik}(\underline{\sigma}_{k \setminus i}) := \frac{1}{|\mathcal{A}_i|} \sum_{\sigma_i \in \mathcal{A}_i} f_k(\underline{\sigma}_k), \qquad (2)$$

where $\underline{\sigma}_{k \setminus i}$ denotes the vector $\underline{\sigma}_k$ without $\sigma_i$, and define the *locally centered basis functions*,

$$g_{ik}(\underline{\sigma}_k) = f_k(\underline{\sigma}_k) - \phi_{ik}(\underline{\sigma}_{k \setminus i}). \qquad (3)$$

As their name suggests, the locally centered basis functions sum to zero $\sum_{\sigma_i \in \mathcal{A}_i} g_{ik}(\underline{\sigma}_k) = 0$. To ensure the scales of the parameters are well defined, we assume that $\theta_k^*$ are chosen or rescaled such that all locally centered basis functions are *normalized* in the following sense:

$$\max_{\underline{\sigma}_k} |g_{ik}(\underline{\sigma}_k)| \le 1, \qquad (4)$$

for all vertices $i \in \mathcal{V}$ and basis factor $k \in \mathcal{K}_i$. This normalization can always be achieved by choosing bounded basis functions $|f_k(\underline{\sigma}_k)| \le 1/2$. An important special case is when the basis functions are already centered, i.e. $g_{ik}(\underline{\sigma}_k) = f_k(\underline{\sigma}_k)$. In this case the basis functions are directly normalized $\max_{\underline{\sigma}_k} |f_k(\underline{\sigma}_k)| = 1$. Note that one of the reasons to define the normalization in (4) in terms of the centered functions $g_k$ instead of $f_k$ is to avoid spurious cases where the functions $f_k$ have inflated magnitudes due to addition of constants $f_k \leftarrow f_k + C$. In Section A of the Supplementary Material, we show that the other important reason to employ centered functions is that degeneracy of the local parameterization with these functions translates to degeneracy of the global distribution in Eq. (1).

## 2.2 Model selection problem

For each $i \in [p]$, let $\mathcal{T}_i \subseteq \mathcal{K}_i$ denote the set of *target factors* that we aim at reconstructing accurately and let $\mathcal{R}_i = \mathcal{K}_i \setminus \mathcal{T}_i$ be the set of *residual factors* for which we do not need learning guarantees. The target and residual parameters are defined similarly as $\underline{\theta}_{\mathcal{T}_i}^* = \{\theta_k^* \mid k \in \mathcal{T}_i\}$ and $\underline{\theta}_{\mathcal{R}_i}^* = \{\theta_k^* \mid k \in \mathcal{R}_i\}$ respectively. Given independent samples from a model in the family in Section 2.1, the goal of the model selection problem is to reconstruct the target parameters of the model.

**Definition 1** (Model Selection Problem). *Given $n$ i.i.d. samples $\underline{\sigma}^{(1)}, \ldots, \underline{\sigma}^{(n)}$ drawn from some distribution $\mu(\underline{\sigma})$ in Eq. (1) defined by $\underline{\theta}^*$, and prior information on $\underline{\theta}^*$ given in form of an upper bound on the $\ell_1$-norm of the local sub-components*

$$\|\underline{\theta}_i^*\|_1 \le \widehat{\gamma}, \qquad (5)$$

*and a* local constraint set $\mathcal{Y}_i \subseteq \mathbb{R}^{\mathbf{K}_i}$ *for each $i \in [p]$ such that*

$$\underline{\theta}_i^* \in \mathcal{Y}_i, \qquad (6)$$

*compute estimates $\widehat{\underline{\theta}}$ of $\underline{\theta}^*$ such that the estimates of the target parameters satisfy*

$$\|\widehat{\underline{\theta}}_{\mathcal{T}_i} - \underline{\theta}_{\mathcal{T}_i}^*\| \le \frac{\alpha}{2}, \quad \forall i \in [p], \qquad (7)$$

*where $\| \cdot \|$ denotes some norm of interest with respect to which the error is measured.*

The bound on the $\ell_1$-norm in (5) is a natural generalization of the sparse case where $\underline{\theta}^*$ only has a small number of non-zero components; in the context of parameter estimation in graphical models, the setting of parameters bounded in the $\ell_1$-norm has been previously considered in [12]. The constraint sets $\mathcal{Y}_i$ are used to encode any other side information that may be known about the model.

## 2.3 Sufficient conditions for well-posedness

We describe some conditions on the model in (1) that makes the model selection problem in Definition 1 well-posed. We first state the conditions formally.

**Condition 1.** *The model from which the samples are drawn in the model selection problem in Definition 1 satisfies the following:*

*(C1)* ***Local Learnability Condition for Graphical Models:*** *There exists a constant $\rho_i > 0$ such that for every vertex $i$ and any vector in the perturbation set $\underline{x} \in \mathcal{X}_i \subseteq \mathbb{R}^{\mathbf{K}_i}$ defined as*

$$\mathcal{X}_i = \{\underline{x} = \underline{y}_1 - \underline{y}_2 \mid \underline{y}_1, \underline{y}_2 \in \mathcal{Y}_i, \|\underline{y}_1\|_1 \le \widehat{\gamma}, \|\underline{y}_2\|_1 \le \widehat{\gamma}\}, \tag{8}$$

*the following holds:*

$$\mathbb{E}\left[\left(\sum_{k \in \mathcal{K}_i} x_k g_{ik}(\underline{\sigma}_k)\right)^2\right] \ge \rho_i \|\underline{x}_{\mathcal{T}_i}\|^2, \tag{9}$$

*where $\underline{x}_{\mathcal{T}_i}$ denotes the components $k \in \mathcal{T}_i$ of $x$, and $\|\cdot\|$ is the norm used in Definition 1.*

*(C2)* ***Finite Maximum Interaction Strength:*** *The following quantity $\gamma$ is finite,*

$$\gamma = \max_{i \in \mathcal{V}} |\max_{\underline{\sigma}} \sum_{k \in \mathcal{K}_i} \theta_k^* g_{ik}(\underline{\sigma}_k)| < \infty. \tag{10}$$

Condition (C1) consists in satisfying the inequality in Eq. (9) involving a quadratic form $\underline{x}^\top \widetilde{I} \underline{x}$ where the matrix $\widetilde{I}$ has indices $k, k' \in \mathcal{K}_i$ and is explicitly defined as $\widetilde{I}_{k,k'} = \mathbb{E}[g_{ik}(\underline{\sigma}_k) g_{ik'}(\underline{\sigma}_{k'})]$. This matrix $\widetilde{I}$ is in fact related to the *conditional* Fisher information matrix.

The conditional Fisher information matrix $I$ with indices $k, k' \in \mathcal{K}_i$ is derived from the conditional distribution of $\sigma_i$ given the remaining variables and reads,

$$I_{k,k'} = \mathbb{E}[g_{ik}(\underline{\sigma}_k) g_{ik'}(\underline{\sigma}_{k'})] - \mathbb{E}_{(\underline{\sigma}_{\backslash i})}\left[\mathbb{E}_{(\sigma_i | \underline{\sigma}_{\backslash i})}[g_{ik}(\underline{\sigma}_k)] \mathbb{E}_{(\sigma_i | \underline{\sigma}_{\backslash i})}[g_{ik'}(\underline{\sigma}_{k'})]\right]. \tag{11}$$

We immediately see that the matrix $\widetilde{I}$ dominates the conditional Fisher information matrix in the positive semi-definite sense, that is $\underline{x}^\top \widetilde{I}(\theta^*) \underline{x} \ge \underline{x}^\top I(\theta^*) \underline{x}$ for all $\underline{x} \in \mathbb{R}^{\mathbf{K}_i}$. Therefore, Condition *(C1)* is satisfied whenever the conditional Fisher information matrix is non-singular in the parameter subspace $\underline{x}_{\mathcal{T}_i}$ that we care to reconstruct and which is compatible with our priors, i.e. for $\underline{x} \in \mathcal{X}_i$. We would like to add that the conditional Fisher information matrix is a natural quantity to consider in this problem as we deliberately focus on using conditional statistics rather than global ones in order to bypass the intractability of the global log-likelihood approach. We are strongly convinced that it should appear in any analysis that entails conditional statistics.

Condition *(C2)* is required to ensure that the model can be recovered with finitely many samples. For many special cases, such as the Ising model, the minimum number of samples required to estimate the parameters must grow exponentially with the maximum interaction strength [17]. A more detailed discussion about well-posedness and Conditions *(C1)* and *(C2)* can be found in Section A of the Supplementary Material.

Conditions *(C1)* and *(C2)* differ from the concepts in [18] called restricted strong convexity property and bound on the interaction strength, respectively, in a subtle but critical manner. Conditions *(C1)* can be identified with restricted strong convexity only when the $\ell_2$-norm is used in Eq. (9). We will see later that the notion of restricted strong convexity is not required for the $\ell_\infty$-norm that appears to be a natural metric for which the local learnability condition can be verified for general models. Moreover, for general models it remains unclear whether the restricted strong convexity holds for values of $\rho_i$ that are independent of the problem dimension $p$. Condition *(C2)* is a weaker assumption than the bound on the interaction strength from [17] for it does not require an extra assumption on the maximum degree of the graphical model.

# 3 Generalized interaction screening

In this Section, we introduce the algorithm that efficiently solves the model selection problem in Definition 1 and provides rigorous guarantees on its reconstruction error and computational complexity.

## 3.1 Generalized regularized interaction screening estimator

We propose a generalization of the estimator RISE, first introduced in [18] for pairwise binary graphical models, in order to reconstruct general discrete graphical models defined in (1). The

*generalized interaction screening objective* (GISO) is defined for each vertex $u$ separately and is given by

$$\mathcal{S}_n(\underline{\theta}_u) = \frac{1}{n} \sum_{t=1}^{n} \exp\left( -\sum_{k \in \mathcal{K}_u} \theta_k g_{uk}(\underline{\sigma}_k^{(t)}) \right), \tag{12}$$

where $\underline{\sigma}^{(1)}, \dots, \underline{\sigma}^{(n)}$ are $n$ i.i.d samples drawn from $\mu(\underline{\sigma})$ in Eq. (1), $\underline{\theta}_u := (\theta_k)_{k \in \mathcal{K}_u}$ is the vector of parameters associated with the factors in $\mathcal{K}_u$ and the locally centered basis functions $g_{uk}$ are defined as in Eq. (3). The GISO retains the main feature of the interaction screening objective (ISO) in [18]: it is proportional to the inverse of the factor in $\mu(\underline{\sigma})$, except for the additional centering terms $\phi_{uk}$. The GISO is a convex function of $\underline{\theta}_u$ and retains the "screening" property of the original ISO. The GISO is used to define the generalized regularized interaction screening estimator (GRISE) for the parameters given by

$$\widehat{\underline{\theta}}_u = \underset{\underline{\theta}_u \in \mathcal{Y}_u : \|\underline{\theta}_u\|_1 \leq \widehat{\gamma}}{\operatorname{argmin}} \mathcal{S}_n(\underline{\theta}_u), \tag{13}$$

where $\widehat{\gamma}$ and $\mathcal{Y}_u$ are the prior information available on $\underline{\theta}_u^*$ as defined in (5) and (6).

## 3.2 Error bound on parameter estimation with GRISE

We now state our first main result regarding the theoretical guarantees on the parameters reconstructed by GRISE. We call $\widehat{\underline{\theta}}_u$ an $\epsilon$-optimal solution of (13) if

$$\mathcal{S}_n(\widehat{\underline{\theta}}_u) \leq \underset{\underline{\theta}_u \in \mathcal{Y}_u : \|\underline{\theta}_u\|_1 \leq \widehat{\gamma}}{\min} \mathcal{S}_n(\underline{\theta}_u) + \epsilon. \tag{14}$$

**Theorem 1** (Error Bound on GRISE Estimates). *Let $\underline{\sigma}^{(1)}, \dots, \underline{\sigma}^{(n)}$ be i.i.d. samples drawn according to $\mu(\underline{\sigma})$ in (1). For some node $u \in \mathcal{V}$, assume that the model satisfies Condition 1 for some norm $\| \cdot \|$ and some constraint set $\mathcal{Y}_u$, and let $\alpha > 0$ be the prescribed accuracy level. If the number of samples satisfies*

$$n \geq 2^{14} \frac{\widehat{\gamma}^2 (1 + \widehat{\gamma})^2 e^{4\gamma}}{\alpha^4 \rho_u^2} \log(\frac{4\mathbf{K}_u^2}{\delta}), \tag{15}$$

*then, with probability at least $1 - \delta$, any estimate that is an $\epsilon$-minimizer of GRISE, with $\epsilon \leq (\rho_u \alpha^2 e^{-\gamma})/(20(1 + \widehat{\gamma}))$, satisfies $\|\widehat{\underline{\theta}}_{\mathcal{T}_u} - \underline{\theta}_{\mathcal{T}_u}^*\| \leq \frac{\alpha}{2}$.*

The proof of Theorem 1 can be found in Section B of the Supplementary Material.

The computational complexity of finding an $\epsilon$-optimal solution of GRISE for a trivial constraint set $\mathcal{Y}_u = \mathbb{R}^{\mathbf{K}_u}$ is $C \frac{c_g n \mathbf{K}_u}{\epsilon^2} \ln(1 + \mathbf{K}_u)$, where $c_g$ is an upper-bound on the computational complexity of evaluating any $g_{ik}(\underline{\sigma}_k)$ for $k \in \mathcal{K}_i$, and $C$ is a universal constant independent of all the parameters of the problem, see Proposition 5 in Section C of the Supplementary Material. For a certain class of constraint sets $\mathcal{Y}_u$, which we term parametrically complete, the problem can be solved in two steps: first, finding a solution to an unconstrained problem, and then projecting onto this set. Note, however, that in general the problem of finding $\epsilon$-optimal solutions to constrained GRISE can still be difficult since the constraint set $\mathcal{Y}_u$ can be arbitrarily complicated.

**Definition 2.** *The constraint set $\mathcal{Y}_u$ is called a parametrically complete set if for all $\underline{\theta}_u \in \mathbb{R}^{|\mathbf{K}_u|}$, there exists $\widehat{\underline{\theta}}_u \in \mathcal{Y}_u$ such that for all $\underline{\sigma}_u$, we have*

$$\sum_{k \in \mathcal{K}_u} \theta_k g_{uk}(\underline{\sigma}_k) = \sum_{k \in \mathcal{K}_u} \widehat{\underline{\theta}}_k g_{uk}(\underline{\sigma}_k). \tag{16}$$

*Any $\widehat{\underline{\theta}}_k \in \mathcal{Y}_u$ satisfying (16) is called an* equi-cost projection *of $\underline{\theta}_u$ onto $\mathcal{Y}_u$ and is denoted by*

$$\widehat{\underline{\theta}}_u \in \mathcal{P}_{\mathcal{Y}_u}(\underline{\theta}_u). \tag{17}$$

The computational complexity of finding of an $\epsilon$-optimal solution of GRISE with parametrically complete set is $C \frac{c_g n \mathbf{K}_u}{\epsilon^2} \ln(1 + \mathbf{K}_u) + \mathcal{C}(\mathcal{P}_{\mathcal{Y}_u}(\widehat{\underline{\theta}}_u^{\text{unc}}))$, where $\mathcal{C}(\mathcal{P}_{\mathcal{Y}_u}(\widehat{\underline{\theta}}_u^{\text{unc}}))$ denotes the computational complexity of the projection step, see Theorem 3 in Section C of the Supplementary Material.

As we will see, for many graphical models it is often possible to explicitly construct parametrically complete sets for which the computational complexity of the projection step $\mathcal{C}(\mathcal{P}_{\mathcal{Y}_u}(\widehat{\underline{\theta}}_u^{\text{unc}}))$ is insignificant compared to the computational complexity of unconstrained GRISE.

# 4 Structure identification and parameter estimation

In this Section we show that the structure of graphical models, which is the collection of maximal subsets of variables that are associated through basis functions, as well as the associated parameters, can be efficiently recovered. The key elements are twofold. First, we prove that for maximal cliques, the Local Learnability Condition (LLC) in *(C1)* can be easily verified and yields a LLC constant independent of the system size. Second, we leverage this property to design an efficient structure and parameter learning algorithm coined SUPRISE that requires iterative calls of GRISE.

## 4.1 The structure of graphical models

The structure plays a central role in graphical model learning for it contains all the information about the conditional independence or Markov property of the distribution $\mu(\underline{\sigma})$ from Eq (1). In order to reach the definition of the structure presented in Eq. (21), we have to introduce graph theoretic concepts specific to graphical models.

The factor graph associated with the model *family* is a bipartite graph $\mathcal{G} = (\mathcal{V}, \mathcal{K}, \mathcal{E})$ with vertex set $\mathcal{V}$, factor set $\mathcal{K}$ and edges connecting factors and vertices,

$$\mathcal{E} = \{(i, k) \subseteq \mathcal{V} \times \mathcal{K} \mid \sigma_i \in \underline{\sigma}_k\}. \tag{18}$$

We see from Eq. (18) that the edge $(i, k)$ exists when the variable $\sigma_i$ associated with the vertex $i$ is an argument of the basis function $f_k(\underline{\sigma}_k)$ associated with the factor $k$. Note that this definition only depends on the set of basis functions $\mathcal{K}$ and does not refer to a particular choice of model within the family. The factor graph $\mathcal{G}^* = (\mathcal{V}, \mathcal{K}^*, \mathcal{E}^*)$ associated with a *model*, as defined in Eq. (1), is the induced subgraph of $\mathcal{G}$ obtained from the vertex set $\mathcal{V}$ and factor subset $\mathcal{K}^* = \{k \in \mathcal{K} \mid \theta_k^* \neq 0\}$. We also use the shorthand notation $\mathcal{G}^* = \mathcal{G}[(\mathcal{V}, \mathcal{K}^*)]$ to denote an induced subgraph of $\mathcal{G}$.

We define the neighbors of a factor $k$ as the set of vertices linked by an edge to $k$ and denote it by $\partial k = \{i \in \mathcal{V} \mid (i, k) \in \mathcal{E}\}$. The largest factor neighborhood size $L = \max_{k \in \mathcal{K}} |\partial k|$ is called the interaction order. Families of graphical models with $L \leq 2$ are referred to as pairwise models as opposed to the generic denomination of $L$-wise models when $L$ is expected to be arbitrary.

The set of maximal factors of a graph is the set of factors whose neighborhood is not strictly contained in the neighborhood of another factor,

$$\mathcal{M}_{\mathrm{fac}}(\mathcal{G}) = \{k \in \mathcal{K} \mid \nexists k' \in \mathcal{K} \text{ s.t } \partial k \subset \partial k'\}. \tag{19}$$

Notice that multiple maximal factors may have the same neighborhood. This motivates the definition of the set of maximal cliques which is contained in the powerset $P(\mathcal{V})$ and consists of all neighborhoods of maximal factors,

$$\mathcal{M}_{\mathrm{cli}}(\mathcal{G}) = \{c \in P(\mathcal{V}) \mid \exists k \in \mathcal{M}_{\mathrm{fac}}(\mathcal{G}) \text{ s.t. } c = \partial k\}. \tag{20}$$

The set of factors whose neighborhoods are the same maximal clique $c \in \mathcal{M}_{\mathrm{cli}}(\mathcal{G})$ is called the span of the clique defined as $[c]_{\mathrm{sp}} = \{k \in \mathcal{M}_{\mathrm{fac}}(\mathcal{G}) \mid c = \partial k\}$. Finally, the structure $\mathbb{S}$ of a graphical model is the set of maximal cliques associated with the factor graph of the *model*,

$$\mathbb{S}(\mathcal{G}^*) = \mathcal{M}_{\mathrm{cli}}(\mathcal{G}^*). \tag{21}$$

We would like to stress that the structure of a model is different from the set of maximal cliques of the *family* of graphical models $\mathcal{M}_{\mathrm{cli}}(\mathcal{G})$ as the former is constructed with the set of factors associated with non-zero parameters while the latter consists of all potential maximal factors.

## 4.2 From local learnability condition to nonsingular parametrization of cliques

We show that the learning problem of reconstructing maximal cliques is well-posed in general and especially for non-degenerate globally centered basis functions. To this end, we demonstrate that the LLC in *(C1)* is automatically satisfied whenever the target sets $\mathcal{T}_i$ consist of factors corresponding to maximal cliques of the graphical model family. Importantly, we prove that the LLC constant $\rho_i$ does not depend on the dimension of the model for the $\ell_{\infty,2}$-norm but rather relies on the Nonsingular Parametrization of Clique (NPC) by the basis functions. Similarly, we also guarantee that the LLC holds for the $\ell_2$-norm in the case of pairwise colorable models.

We introduce globally centered basis functions defined for any factor $k \in \mathcal{K}$ through the inclusion–exclusion formula,

$$h_k\left(\underline{\sigma}_k\right) = f_k(\underline{\sigma}_k) + \sum_{r \in P(\partial k)\setminus\emptyset} \frac{(-1)^{|r|}}{|\mathcal{A}_r|} \sum_{\underline{\sigma}_r} f_k(\underline{\sigma}_k), \tag{22}$$

where $\mathcal{A}_r = \bigotimes_{j \in r} \mathcal{A}_j$. It is straightforward to see that globally centered functions sum partially to zero for any variables, i.e. $\sum_{\sigma_i \in \mathcal{A}_i} h_k(\underline{\sigma}_k) = 0$ for all $i \in \partial k$. It is worth noticing that when the functions $f_k$ are already globally centered, we have $f = g = h$. We would also like to point out that unlike locally centered functions $g_{ik}$, globally centered functions cannot in general be interchanged with functions $f_k$ without modifying conditional distributions. However they play an important role in determining the independence of basis functions around cliques through the Nonsingular Parametrization of Cliques (NPC) constant introduced below. Given a perturbation set $\mathcal{X}_i$, as defined in Eq. (8), the NPC constant is defined through the following minimization,

$$\rho_i^{\mathrm{NPC}} = \min_{\substack{c \in \mathcal{M}_{\mathrm{cli}}(\mathcal{G}) \\ c \ni i}} \min_{\substack{\|\underline{x}_c\|_2=1 \\ \underline{x}_c \in \mathcal{X}_i^c}} \mathbb{E}_{(\sigma_i)} \left[ \sum_{\underline{\sigma}_{c\setminus i} \in \mathcal{A}_{c\setminus i}} \left( \sum_{k \in [c]_{\mathrm{sp}}} x_k h_k(\underline{\sigma}_k) \right)^2 \right], \tag{23}$$

where the vector $\underline{x}_c \in \mathbb{R}^{|[c]_{\mathrm{sp}}|}$ belongs to $\mathcal{X}_i^c$, the projection of the constraint $\mathcal{X}_i \subseteq \mathbb{R}^{\mathbf{K}_i}$ to the components $k \in [c]_{\mathrm{sp}}$ and the expectation is with respect to the marginal distribution of $\sigma_i$. Note that NPC constant only depends on $L$ and not on the size of the system, and can be explicitly computed in time $O(\mathbf{K})$. A detailed discussion can be found in Section D of the Supplementary Material. The importance of the NPC constant is highlighted by the following proposition that guarantees that the LLC is satisfied for maximal factors in $\ell_{\infty,2}$-norm as long as $\rho_i^{\mathrm{NPC}} > 0$.

**Proposition 1** (LLC in $\ell_{\infty,2}$-norm). *For a specific vertex $i \in \mathcal{V}$, let the target set be maximal factors with $i$ as neighbor, i.e. $\mathcal{T}_i = \{k \in \mathcal{M}_{\mathrm{fac}}(\mathcal{G}) \mid \partial k \ni i\}$. For vectors in the perturbation set $\underline{x} \in \mathcal{X}_i \subseteq \mathbb{R}^{\mathbf{K}_i}$, define the $\ell_{\infty,2}$-norm over components that are maximal factors as $\|\underline{x}_{\mathcal{T}_i}\|_{\infty,2} = \max_{\substack{c \in \mathcal{M}_{\mathrm{cli}}(\mathcal{G}) \\ c \ni i}} \sqrt{\sum_{k \in [c]_{\mathrm{sp}}} x_k^2}$. Then for discrete graphical models with maximum alphabet size $q$, interaction order $L$ and models with finite maximum interaction strength $\gamma$ as defined in Eq. (10), the Local Learnability Condition (C1) is satisfied whenever the Nonsingular Parameterization of Cliques constant $\rho_i^{\mathrm{NPC}}$ is nonzero and we have,*

$$\mathbb{E}\left[ \left( \sum_{k \in \mathcal{K}_i} x_k g_{ik}(\underline{\sigma}_k) \right)^2 \right] \geq \rho_i^{\mathrm{NPC}} \left( \frac{\exp(-2\gamma)}{q} \right)^{L-1} \|\underline{x}_{\mathcal{T}_i}\|_{\infty,2}^2. \tag{24}$$

Proposition 1 guarantees that the LLC can be satisfied uniformly in the size $p$ of the model whenever $\rho_i^{\mathrm{NPC}} > 0$. The proof of Proposition 1 can be found in Section D of the Supplementary Material.

For family of models whose factors involve at most $L = 2$ variables, the so-called pairwise models, we can show that the LLC conditions for maximal factors also holds for the $\ell_2$-norm. This LLC conditions depends on the vertex chromatic number $\chi$ of the *model* factor graph. We recall that a vertex coloring of a graph $\mathcal{G}^* = (\mathcal{V}, \mathcal{K}^*, \mathcal{E}^*)$ is a partition $\{S_r\}_{r \in \mathbb{N}} \in P(\mathcal{V})$ of the vertex set such that no two vertices with the same color are connected to the same factor node, i.e. $i, j \in S_r \Rightarrow \nexists k \in \mathcal{K}^*$ s.t. $i, j \in \partial k$. The chromatic number is the cardinality of the smallest graph coloring.

**Proposition 2** (LLC in $\ell_2$-norm for pairwise models). *For a specific vertex $i \in \mathcal{V}$, let the target set be maximal factors with $i$ as neighbor, i.e. $\mathcal{T}_i = \{k \in \mathcal{M}_{\mathrm{fac}}(\mathcal{G}) \mid \partial k \ni i\}$. For $\underline{x} \in \mathcal{X}_i \subseteq \mathbb{R}^{\mathbf{K}_i}$, define the $\ell_2$-norm over components that are maximal factors $\|\underline{x}_{\mathcal{T}_i}\|_2 = \sqrt{\sum_{k \in \mathcal{T}_i} x_k^2}$. Then for discrete pairwise graphical models with maximum alphabet size $q$ and models with chromatic number $\chi$ and finite maximum interaction strength $\gamma$ as defined in Eq. (10), the Local Learnability Condition (C1) is satisfied whenever the NPC constant $\rho_i^{\mathrm{NPC}}$ is nonzero and we have,*

$$\mathbb{E}\left[ \left( \sum_{k \in \mathcal{K}_i} x_k g_{ik}(\underline{\sigma}_k) \right)^2 \right] \geq \frac{\rho_i^{\mathrm{NPC}}}{\chi} \frac{\exp(-2\gamma)}{q} \|\underline{x}_{\mathcal{T}_i}\|_2^2. \tag{25}$$

The reader will find the proof of Proposition 2 in Section D of the Supplementary Material.

### 4.3 Structure unveiling and parameter reconstruction with interaction screening estimation

---

**Algorithm 1:** Structure Unveiling and Parameter Reconstruction with Interaction Screening Estimation (SUPRISE)

---

```
// Step 1: Initialization of set of considered factors
```
1  $\mathcal{K}^0 \leftarrow \mathcal{K}$ ;
2  **for** $t = 0, \ldots, L - 1$ **do**
```
        // Step 2: Reconstruct maximal factors bigger than L − t
```
3      Construct the induced sub-graph: $\mathcal{G}^t \leftarrow \mathcal{G}\left[(\mathcal{V}, \mathcal{K}^t)\right]$;
4      **for** $u \in \mathcal{V}$ **do**
5          Set target factors: $\mathcal{T}_u^t \leftarrow \{k \in \mathcal{M}_{\text{fac}}\left(\mathcal{G}^t\right) \mid \partial k \ni u\}$;
6          Set residual factors: $\mathcal{R}_u^t \leftarrow \mathcal{K}_u^t \setminus \mathcal{T}_u^t$;
7          Estimate $\widehat{\underline{\theta}}_u^t$ using GRISE with accuracy at least
           $\epsilon = \rho_{\text{NPC}}\alpha^2 \exp(-\gamma(2L - 1))/(20(1 + \widehat{\gamma})q^{L-1})$ on the model defined by $\mathcal{K}_u^t, \mathcal{T}_u^t, \mathcal{R}_u^t$
           and constraint set $\mathcal{Y}_u$;
8      **end**
```
        // Step 3: Identify max cliques associated with zero parameters
```
9      Initialize set of removable factors: $\mathcal{N}^t \leftarrow \emptyset$;
10     **for** $c \in \mathcal{M}_{\text{cli}}\left(\mathcal{G}^t\right)$ **do**
11         Compute average reconstruction: $\widehat{\underline{\theta}}_c^{\text{avg}(t)} \leftarrow \left\{ |c|^{-1} \sum_{u \in c} (\widehat{\underline{\theta}}_u^t)_k \mid k \in [c]_{\text{sp}} \right\}$;
12         **if** $\|\widehat{\underline{\theta}}_c^{\text{avg}(t)}\|_2 < \alpha/2$ **then**
13            Update set of removable factors: $\mathcal{N}^t \leftarrow \mathcal{N}^t \cup [c]_{\text{sp}}$ ;
14         **end**
15     **end**
16     Update considered factors: $\mathcal{K}^{t+1} \leftarrow \mathcal{K}^t \setminus \mathcal{N}^t$;
17 **end**
```
// Step 4: output structure and non-zero parameters of maximal factors
```
18 **return** $\widehat{\mathbb{S}} = \mathcal{M}_{\text{cli}}\left(\mathcal{G}\left[(\mathcal{V}, \mathcal{K}^L)\right]\right)$ and $\widehat{\underline{\theta}}_{\mathcal{M}} = \left\{ \widehat{\theta}_k^{\text{avg}(L-1)} \mid k \in \mathcal{M}_{\text{fac}}\left(\mathcal{G}\left[(\mathcal{V}, \mathcal{K}^L)\right]\right) \right\}$;

---

Suppose that we know $\alpha > 0$, a lower-bound on the minimal intensity of the parameters associated with the structure in the sense that $\alpha \le \min_{c \in \mathbb{S}(\mathcal{G}^*)} \sqrt{\sum_{k \in [c]_{\text{sp}}} \theta_k^{*2}}$. Then we can recover the structure and parameters associated with maximal factors of any graphical models using Algorithm 1, coined SUPRISE for Structure Unveiling and Parameter Reconstruction with Interaction Screening Estimation. SUPRISE that implements an iterative use of GRISE is shown to have a sample complexity logarithmic in the system size for models with non-zero NPC constants. Our second main result is the following Theorem 2, proved in Section D of the Supplementary Material, which provides guarantees on SUPRISE.

**Theorem 2 (Reconstruction and Estimation Guarantees for SUPRISE).** *Let $\mu(\underline{\sigma})$ in (1) be the probability distribution of a discrete graphical model with maximum alphabet size $q$, interaction order $L$, finite maximum interaction strength $\gamma$ and smallest Nonsingular Parameterization of Cliques constant greater than zero, i.e. $\rho_{\text{NPC}} = \min_{u \in \mathcal{V}} \rho_u^{\text{NPC}} > 0$. Let $\underline{\sigma}^{(1)}, \ldots, \underline{\sigma}^{(n)}$ be i.i.d. samples drawn according to $\mu(\underline{\sigma})$ and assume that*

$$n \ge 2^{14} q^{2(L-1)} \frac{\widehat{\gamma}^2 (1 + \widehat{\gamma})^2 e^{4\gamma L}}{\alpha^4 \rho_{\text{NPC}}^2} \log\left(\frac{4pL\mathbf{K}^2}{\delta}\right), \tag{26}$$

*where $\mathbf{K} = \max_{u \in \mathcal{V}} \mathbf{K}_u$ is the maximal number of basis functions in which a variable can appear and $\widehat{\gamma} \ge \max_{u \in \mathcal{V}} \|\underline{\theta}_u^*\|_1$ is our $\ell_1$-prior on the components of the parameters. Then the structure of the general graphical model is perfectly recovered using Algorithm 1, i.e. $\widehat{\mathbb{S}} = \mathbb{S}$, with probability $1 - \delta$. In addition, the parameters associated with maximal factors are reconstructed with precision $\max_{c \in \mathbb{S}} \sum_{k \in [c]_{\text{sp}}} \left(\widehat{\theta}_k - \theta_k^*\right)^2 \le \alpha^2/4$ for general models and with $\sum_{c \in \mathbb{S}} \sum_{k \in [c]_{\text{sp}}} \left(\widehat{\theta}_k - \theta_k^*\right)^2 \le \chi^2 \alpha^2/4$ for pairwise models with chromatic number $\chi$.*

Table 1: Sample complexity and computational complexity of SUPRISE over special cases.

| Model name | Inter. order | Alphabet size | Recovery type | Sample complexity | Algo. complexity |
|:---:|:---:|:---:|:---:|:---:|:---:|
| Ising | 2 | 2 | structure | $O\left(\alpha^{-4}e^{8\gamma}\log p\right)$ | $\widetilde{O}(p^2)$ |
| Ising | 2 | 2 | $\ell_2$-parameter | $O\left(\chi^2\alpha^{-4}e^{8\gamma}\log p\right)$ | $\widetilde{O}(p^2)$ |
| Binary | $L$ | 2 | structure | $O\left(\alpha^{-4}4^L e^{4\gamma L}L\log p\right)$ | $\widetilde{O}(p^L)$ |
| Pairwise | 2 | $q$ | structure | $O\left(\alpha^{-4}q^4 e^{12\gamma}\log(pq)\right)$ | $\widetilde{O}(p^2)$ |
| Pairwise | 2 | $q$ | $\ell_2$-parameter | $O\left(\chi^2\alpha^{-4}q^4 e^{12\gamma}\log(pq)\right)$ | $\widetilde{O}(p^2)$ |
| General | $L$ | $q$ | structure | $O\left(\alpha^{-4}q^{2L}e^{4\gamma(L+1)}L\log(pq)\right)$ | $\widetilde{O}(p^L)$ |

*The total computational complexity scales as $\widetilde{O}(p\mathbf{K})$, for fixed $L$, $\alpha$, $\gamma$, $\widehat{\gamma}$ and $\delta$, if the constraint sets $\mathcal{Y}_u$ are parametrically complete.*

As an application of Theorem 2, we show the sample and computational complexity of recovering parameter values and the structure of some well-known special cases in Table 1. The parameter $\alpha$ appearing in Table 1 is the precision to which parameters are recovered in the considered norm, $\chi$ is the chromatic number of the graph, $L$ is the interaction order, $q$ is the maximum alphabet size, $\gamma$ is the maximum interaction strength and $p$ is the number of variables. At this point, it is instructive to compare our sample complexity requirements to existing results. A direct application of bounds of [12] and [20] to the case of pairwise multi-alphabet models that we consider below yields $O(\exp(14\gamma))$ dependence, whereas SUPRISE has a complexity that scales as $O(\exp(12\gamma))$. In the case of binary $L$-wise models, while [12] shows the $O(\exp(O(\gamma L)))$ scaling, SUPRISE enjoys a sample complexity $O(\exp(4\gamma L))$. The algorithm of [9] recovers a subclass of general graphical models with bounded degree, but has a sub-optimal double-exponential scaling in $\gamma$, while SUPRISE leads to recovery of arbitrary discrete graphical models with a single-exponential dependence in $\gamma$ and needs no bound on the degree. In terms of the computational complexity, SUPRISE achieves the efficient scaling $\widetilde{O}(p^L)$ for models with the maximum interaction order $L$, which matches the best-known $\widetilde{O}(p^2)$ scaling for pairwise models [12, 20]. In summary, SUPRISE generalizes, improves and extends the existing results in the literature. The proofs for special cases can be found in Section E of the Supplementary Material.

## 5 Conclusion and future work

A key result of our paper is the existence of a computationally efficient algorithm that is able to recover arbitrary discrete graphical models with multi-body interactions. This result is a particular case of the general framework that we have introduced, which considers arbitrary model parametrization and makes distinction between the bounds on the parameters of the underlying model and the prior parameters. The computational complexity $\widetilde{O}(p^L)$ that we achieve is believed to be efficient for this problem [12]. In terms of sample complexity, the information-theoretic bounds for recovery of general discrete graphical models are unknown. In the case of binary pairwise models, the sample complexity bounds resulting from our general analysis are near-optimal with respect to known information-theoretic lower bounds [17]. It would be interesting to see if the $1/\alpha^4$ factor in our sample complexity bounds can be improved to $1/\alpha^2$ using an $\ell_1$-norm penalty rather than an $\ell_1$-norm constraint, as it has been shown for the particular case of Ising models [13, 18].

Other open questions left for future exploration include the possibility to extend the analysis to the case of graphical models with nonlinear parametrizations like in [10], and to graphical models with continuous variables. It is particularly interesting to see whether the computationally efficient and nearly sample-optimal method introduced in the present work could be useful for designing efficient learning algorithms that can improve the state-of-the-art in the well-studied case of Gaussian graphical models, for which it has been recently shown that the information-theoretic lower bound on sample complexity is tight [14].

# Broader impact

We believe that this work, as presented here, does not present any foreseeable societal consequence.

# Acknowledgments

Research presented in this article was supported by the Laboratory Directed Research and Development program of Los Alamos National Laboratory under project numbers 20190059DR, 20190195ER, 20190351ER, and 20210078DR.

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
