[Supplementary Material]

## Supplementary Material

This document contains supplementary materials for the paper "Efficient Learning of Discrete Graphical Models". Section A contains a detailed discussion about Conditions *(C1)* and *(C2)*. In Section B, the reader can find the proofs for the error bound on GRISE estimates. Section C contains the proofs relatives to the computational complexity of GRISE and its efficient implementation. In Section D, the reader can find the proofs in connection with the NPC constant and relative to structure and parameter estimation with SUPRISE. Finally, Section E contains the proofs for the applications of SUPRISE for structure and parameter estimation of iconic special cases.

## A    About well-posedness and local learnability conditions

### A.1    About condition *(C1)*

To illustrate further why Condition *(C1)* is required, we look at a case where the local constraint set is trivial, i.e. $\mathcal{Y}_i = \mathbb{R}^{\mathbf{K}_i}$ and we consider a model that violates Condition *(C1)*. This implies that there exists a sequence $\underline{x}_n \in \mathcal{X}_i$ such that $\underline{x}_n^\top \widetilde{I}(\theta^*)\underline{x}_n / \|\underline{x}_{\mathcal{T}_i}\|^2 < \rho_n$ with $\rho_n \to 0$. In the limit, we can find a vector $\underline{x}$ such that $\underline{x}^\top \widetilde{I}(\theta^*)\underline{x} = 0$ and $\|\underline{x}_{\mathcal{T}_i}\| = 1$. In other words, it implies that for this model there exists a vertex $i$ and a perturbation vector $\underline{x} \in \mathcal{X}_i$ such that $\underline{x}_{\mathcal{T}_i} \neq 0$ and for which $\mathbb{E}\left[\left(\sum_{k \in \mathcal{K}_i} x_k g_{ik}(\underline{\sigma}_k)\right)^2\right] = 0$. Since the probability distribution in Eq. (1) is positive, it further implies that for all configurations $\underline{\sigma}$ we have the functional equality $\sum_{k \in \mathcal{K}_i} x_k g_{ik}(\underline{\sigma}_k) = 0$. This enables us to locally reparameterize the distribution:

$$\exp\left(\sum_{k \in \mathcal{K}_i} \theta_k^* f_k(\underline{\sigma}_k)\right) = \exp\left(c(\underline{x}, \underline{\sigma}_{\backslash i}) + \sum_{k \in \mathcal{K}_i} (\theta_k^* - x_k) f_k(\underline{\sigma}_k)\right), \tag{27}$$

where $c(\underline{x}, \underline{\sigma}_{\backslash i}) = \sum_{k \in \mathcal{K}_i} x_k \phi_{ik}(\underline{\sigma}_{k \backslash i})$ is a sum of locally centered functions that does not involve the variable $\sigma_i$. At this point, we should distinguish between the two cases when the basis functions $f_k$ are centered or not. When the basis functions are centered, i.e. $f_k = g_k$, the residual in Eq. (27) is identically zero, $c(\underline{x}, \underline{\sigma}_{\backslash i}) = 0$. Therefore, the probability distribution of the model in (1) can be reparameterized entirely with $\theta_k^* \to \theta_k^* - x_k$ for $k \in \mathcal{K}_i$. It implies, as $\underline{x}_{\mathcal{T}_i} \neq 0$, that there exists two parameterization of the same models with different target parameters and the model selection problem as stated in Definition 1 is ill-posed. In the case when the basis functions are not centered, i.e. $f_k \neq g_k$, it may not be possible to reparameterized the whole distribution of the model. However, the conditional probability distribution $\mathbb{P}(\sigma_i \mid \underline{\sigma}_{\backslash i})$ can be reparameterized as it is proportional to $\exp\left(\sum_{k \in \mathcal{K}_i} \theta_k^* f_k(\underline{\sigma}_k)\right)$ and $\exp\left(\sum_{k \in \mathcal{K}_i} (\theta_k^* - x_k) f_k(\underline{\sigma}_k)\right)$ thanks to Eq. (27). Thereby, even if the model is uniquely parameterized with $\underline{\theta}_{\mathcal{T}_i}$, local methods based on independent neighborhood reconstructions using conditional distributions will fail at selecting a unique model as shown in the following example.

**Example 1.** *Consider two family of models over two binary variables $\sigma, s \in \{-1, 1\}$, parameterized by $\underline{\theta}$ and $\underline{\eta}$,*

$$\mu_{\underline{\theta}}(\sigma, s) \propto \exp\left(\theta_1 \sigma(s-1) + \theta_2 s(\sigma - 1) + \theta_3(\sigma + s)\right), \tag{28}$$

*and*

$$\mu_{\underline{\eta}}(\sigma, s) \propto \exp\left(\eta_1 \sigma s + \eta_2 \sigma + \eta_3 s\right). \tag{29}$$

*Both models are equivalent through the invertible mapping $\eta_1 = \theta_1 + \theta_2$, $\eta_2 = \theta_3 - \theta_1$ and $\eta_3 = \theta_3 - \theta_2$. However the model in Eq. (29) that has centered basis functions satisfies* (C1) *from Condition 1, while the model in Eq. (28) that has non-centered basis functions does not. This implies that the parameters $\underline{\theta}$ cannot be recovered by looking independently at conditional distributions as they are degenerate in this basis,*

$$\mathbb{P}(\sigma \mid s) \propto \exp\left((\theta_1 + \theta_2)\sigma s + (\theta_3 - \theta_1)\sigma\right), \tag{30}$$

$$\mathbb{P}(s \mid \sigma) \propto \exp\left((\theta_1 + \theta_2)\sigma s + (\theta_3 - \theta_2)s\right). \tag{31}$$

*Indeed the change of parameters $\theta_1 \to \theta_1 + \epsilon$, $\theta_2 \to \theta_2 - \epsilon$ and $\theta_3 \to \theta_3 + \epsilon$ leaves the conditional distribution* (30) *unchanged while the change of parameters $\theta_1 \to \theta_1 - \epsilon$, $\theta_2 \to \theta_2 + \epsilon$ and*

$\theta_3 \to \theta_3 - \epsilon$ *leaves the conditional distribution* (31) *unaffected. Note that there does not exist a change of parameters that leaves both* (30) *and* (31) *unchanged. This is in agreement with the fact that the model in Eq.* (28) *is uniquely parameterized and can be in principle recovered by looking* jointly *at both conditional distributions.*

For the specific models that we considered in Section E, the basis functions are always centered, which implies that failure to satisfy *(C1)* means that the model selection problem is ill-posed.

## A.2 About condition *(C2)*

The bound on the interaction strength in *(C2)* translates directly into a uniform bound on the conditional probabilities of the models as shown in the following lemma.

**Lemma 1** (Lower-Bounded Conditional Probabilities). *Consider a graphical model with bounded maximum interaction strength of $\gamma = \max_{i \in \mathcal{V}} | \max_{\underline{\sigma}} \sum_{k \in \mathcal{K}_i} \theta_k^* g_{ik}(\underline{\sigma}_k)|$. Then for any two disjoint subsets of vertices $A, B \subseteq \mathcal{V}$ the conditional probability of $\underline{\sigma}_A$ given $\underline{\sigma}_B$ is bounded from below,*

$$\mathbb{P}(\underline{\sigma}_A \mid \underline{\sigma}_B) \geq \prod_{i \in A} \frac{\exp(-2\gamma)}{|\mathcal{A}_i|}, \tag{32}$$

*where $|\mathcal{A}_i|$ is the alphabet size of $\sigma_i$.*

*Proof of Lemma 1: Lower-bound on conditional probabilities.* We start by bounding the conditional probability of one variable $\sigma_i$ given the rest $\underline{\sigma}_{\backslash i}$. This is given by the following expression,

$$\mathbb{P}(\sigma_i \mid \underline{\sigma}_{\backslash i}) = \frac{\exp\left(\sum_{k \in \mathcal{K}_i} \theta_k^* f_k(\underline{\sigma}_k)\right)}{\sum_{\sigma_i \in \mathcal{A}_i} \exp\left(\sum_{k \in \mathcal{K}_i} \theta_k^* f_k(\underline{\sigma}_k)\right)}, \tag{33}$$

$$= \frac{\exp\left(\sum_{k \in \mathcal{K}_i} \theta_k^* \left(g_{ik}(\underline{\sigma}_k) + \phi_{ik}(\sigma_{k \backslash i})\right)\right)}{\sum_{\sigma_i \in \mathcal{A}_i} \exp\left(\sum_{k \in \mathcal{K}_i} \theta_k^* \left(g_{ik}(\underline{\sigma}_k) + \phi_{ik}(\sigma_{k \backslash i})\right)\right)}, \tag{34}$$

$$= \frac{\exp\left(\sum_{k \in \mathcal{K}_i} \theta_k^* g_{ik}(\underline{\sigma}_k)\right)}{\sum_{\sigma_i \in \mathcal{A}_i} \exp\left(\sum_{k \in \mathcal{K}_i} \theta_k^* g_{ik}(\underline{\sigma}_k)\right)}, \tag{35}$$

as the centering functions $\phi_{ik}(\sigma_{k \backslash i})$ are independent of $\sigma_i$. The last expression can be simply bounded away from zero,

$$\mathbb{P}(\sigma_i \mid \underline{\sigma}_{\backslash i}) = \frac{\exp\left(\sum_{k \in \mathcal{K}_i} \theta_k^* g_{ik}(\underline{\sigma}_k)\right)}{\sum_{\sigma_i \in \mathcal{A}_i} \exp\left(\sum_{k \in \mathcal{K}_i} \theta_k^* g_{ik}(\underline{\sigma}_k)\right)} \geq \frac{\exp(-2\gamma)}{|\mathcal{A}_i|}, \tag{36}$$

using $\gamma \geq |\sum_{k \in \mathcal{K}_i} \theta_k^* g_{ik}(\underline{\sigma}_k)|$.

Now we consider the conditional probability of one variable $\sigma_i$ given a subset of variable $\underline{\sigma}_B$ where $B \subseteq \mathcal{V}$ and $i \notin B$. Denote the complementary set of $i$ and $B$ by $S = \mathcal{V} \setminus (\{i\} \cup B)$. Then using the chain rule and the inequality from Eq. (36) we find the following lower-bound,

$$\mathbb{P}(\sigma_i \mid \underline{\sigma}_B) = \sum_{\underline{\sigma}_S} \mathbb{P}(\sigma_i, \underline{\sigma}_S \mid \underline{\sigma}_B) \tag{37}$$

$$= \sum_{\underline{\sigma}_S} \mathbb{P}(\sigma_i \mid \underline{\sigma}_S, \underline{\sigma}_B) \mathbb{P}(\underline{\sigma}_S \mid \underline{\sigma}_B) \tag{38}$$

$$= \sum_{\underline{\sigma}_S} \mathbb{P}(\sigma_i \mid \underline{\sigma}_{\backslash i}) \mathbb{P}(\underline{\sigma}_S \mid \underline{\sigma}_B) \tag{39}$$

$$\geq \frac{\exp(-2\gamma)}{|\mathcal{A}_i|}. \tag{40}$$

Finally we consider the conditional probability of a set of variable $\underline{\sigma}_A$ given another set $\underline{\sigma}_B$ where $A \cap B = \emptyset$. Denote the vertices in $A$ by $\{1, 2, \ldots, |A|\}$. Then using the chain rule and the inequality

from Eq. (36), we obtain the desired result,

$$\mathbb{P}(\underline{\sigma}_A \mid \underline{\sigma}_B) = \prod_{j=1,\dots,|A|} \mathbb{P}(\sigma_j, \mid \underline{\sigma}_B, \sigma_{j+1}, \dots, \sigma_{|A|}), \tag{41}$$

$$\geq \prod_{j=1,\dots,|A|} \frac{\exp(-2\gamma)}{|\mathcal{A}_j|}. \tag{42}$$

□

## B   Proofs of GRISE estimation error bound

**Proposition 3** (Gradient Concentration for GRISE). *For some node $u \in V$, let $n > \frac{2e^{2\gamma}}{\epsilon_1^2} \log(\frac{2\mathbf{K}_u}{\delta_1})$, then with probability at least $1 - \delta_1$ the components of the gradient of the GISO are bounded from above as*

$$\|\nabla \mathcal{S}_n(\underline{\theta}_u^*)\|_\infty < \epsilon_1. \tag{43}$$

Define the residual of the first order Taylor expansion as

$$\delta \mathcal{S}_n(\Delta, \underline{\theta}_u^*) = \mathcal{S}_n(\underline{\theta}_u^* + \Delta) - \mathcal{S}_n(\underline{\theta}_u^*) - \langle \nabla \mathcal{S}_n(\underline{\theta}_u^*), \Delta \rangle. \tag{44}$$

**Proposition 4** (Restricted Strong Convexity for GRISE). *For some node $u \in \mathcal{V}$, let $n > \frac{2}{\epsilon_2^2} \log\left(\frac{2\mathbf{K}_u^2}{\delta_2}\right)$ and assume that Condition 1 holds for some norm $\|\cdot\|$. Then, with probability at least $1 - \delta_2$ the error of the first order Taylor expansion of the GISO satisfies*

$$\delta \mathcal{S}_n(\Delta, \underline{\theta}_u^*) \geq \exp(-\gamma) \frac{\rho_u \|\Delta_{\mathcal{T}_u}\|^2 - \epsilon_2 \|\Delta\|_1^2}{2 + \|\Delta\|_1}. \tag{45}$$

*for all $\Delta \in \mathcal{X}_u \subseteq \mathbb{R}^{\mathbf{K}_u}$.*

We first prove Theorem 1 before proving the propositions.

*Proof of Theorem 1: Error Bound on GRISE.* For some node $u \in \mathcal{V}$, let $n \geq \frac{2^{14}\widehat{\gamma}^2(1+\widehat{\gamma})^2 e^{4\gamma}}{\alpha^4 \rho_u^2} \log(\frac{4\mathbf{K}_u^2}{\delta})$. As the estimate $\widehat{\underline{\theta}}_u$ is an $\epsilon$-optimal point of the GISO and $\underline{\theta}_u^*$ lies in the constraint set from Eq. (6), we find that for $\Delta = \widehat{\underline{\theta}}_u - \underline{\theta}_u^*$

$$\epsilon \geq \mathcal{S}_n(\widehat{\underline{\theta}}_u) - \mathcal{S}_n(\underline{\theta}_u^*) \tag{46}$$

$$= \langle \nabla \mathcal{S}_n(\underline{\theta}_u^*), \Delta \rangle + \delta \mathcal{S}_n(\Delta, \underline{\theta}_u^*) \tag{47}$$

$$\geq -\|\nabla \mathcal{S}_n(\underline{\theta}_u^*)\|_\infty \|\Delta\|_1 + \delta \mathcal{S}_n(\Delta, \underline{\theta}_u^*). \tag{48}$$

Using the union bound on Proposition 3 and Proposition 4 with $\delta_1 = \delta_2 = \frac{\delta}{2}$ and

$$\epsilon \leq \frac{\rho_u \alpha^2 e^{-\gamma}}{20(1+\gamma)}, \epsilon_1 = \frac{\rho_u \alpha^2 e^{-\gamma}}{40\widehat{\gamma}(1+\widehat{\gamma})}, \epsilon_2 = \frac{\rho_u \alpha^2}{80\widehat{\gamma}^2}, \tag{49}$$

we can express the inequality as

$$\epsilon \geq -\epsilon_1 \|\Delta\|_1 + e^{-\gamma} \frac{\rho_u \|\Delta_{\mathcal{T}_u}\|^2 - \epsilon_2 \|\Delta\|_1^2}{2 + \|\Delta\|_1}. \tag{50}$$

Since by assumptions $\|\underline{\theta}_u^*\|_1 \leq \gamma$ and $\|\widehat{\underline{\theta}}_u\|_1 \leq \widehat{\gamma}$ for $\gamma \leq \widehat{\gamma}$ as the estimate is an $\epsilon$-optimal point of the $\ell_1$-*constrained* GISO, the error $\|\Delta\|_1$ is bounded by $2\widehat{\gamma}$. By choosing

and after some algebra, we obtain that

$$\|\Delta_{\mathcal{T}_u}\| \leq \frac{\alpha}{2}. \tag{51}$$

□

## B.1 Gradient concentration

The components of the gradient of the GISO is given by

$$\frac{\partial}{\partial \theta_k} \mathcal{S}_n(\underline{\theta}_u^*) = \frac{1}{n} \sum_{t=1}^{n} -g_{uk}(\underline{\sigma}_k^{(t)}) \exp\left(-\sum_{l \in \mathcal{K}_u} \theta_l^* g_{ul}(\underline{\sigma}_l^{(t)})\right). \tag{52}$$

Each term in the summation above is distributed as the random variable

$$X_{uk} = -g_{uk}(\underline{\sigma}_k) \exp\left(-\sum_{l \in \mathcal{K}_u} \theta_k^* g_{ul}(\underline{\sigma}_l)\right) \quad \forall k \in \mathcal{K}_u. \tag{53}$$

**Lemma 2.** *For any $u \in \mathcal{V}$ and $k \in \mathcal{K}_u$, we have*

$$\mathbb{E}\left[X_{uk}\right] = 0. \tag{54}$$

*Proof.* Simple computation. $\qquad\square$

*Proof of Proposition 3: Gradient Concentration for* GRISE. The random variable $X_{uk}$ is bounded as

$$|X_{uk}| = |g_{uk}(\underline{\sigma}_k)| \exp\left(-\sum_{k \in \mathcal{K}_u} \theta_k^* g_{uk}(\underline{\sigma}_k)\right) \leq \exp(\gamma). \tag{55}$$

Using Lemma 2 and the Hoeffding inequality, we get

$$\mathbb{P}\left(\left|\frac{\partial}{\partial \theta_k} \mathcal{S}_n(\underline{\theta}_u^*)\right| > \epsilon_1\right) < 2\exp\left(-\frac{n\epsilon_1^2}{2e^{2\gamma}}\right). \tag{56}$$

The proof follows by using (56) and the union bound over all $k \in \mathcal{K}_u$. $\qquad\square$

## B.2 Restricted strong convexity

We make use of the following deterministic functional inequality derived in [18].

**Lemma 3.** *The following inequality holds for all $z \in \mathbb{R}$.*

$$e^{-z} - 1 + z \geq \frac{z^2}{2 + |z|}. \tag{57}$$

*Proof of Lemma 3.* Note that the inequality is true for $z = 0$ and the first derivative of the difference is positive for $z > 0$ and negative for $z < 0$. $\qquad\square$

Let $H_{k_1 k_2}$ denote the correlation between $g_{k_1}$ and $g_{k_2}$ defined as

$$H_{k_1 k_2} = \mathbb{E}\left[g_{uk_1}(\underline{\sigma}_{k_1}) g_{uk_2}(\underline{\sigma}_{k_2})\right], \tag{58}$$

and let $H = [H_{k_1 k_2}] \in \mathbb{R}^{|\mathcal{K}_u| \times |\mathcal{K}_u|}$ be the corresponding matrix. We define $\hat{H}$ similarly based on the empirical estimates of the correlation $\hat{H}_{k_1 k_2} = \frac{1}{n} \sum_{t=1}^{n} g_{uk_1}(\underline{\sigma}_{k_1}^{(t)}) g_{uk_2}(\underline{\sigma}_{k_2}^{(t)})$. The following lemma bounds the deviation between the above two quantities.

**Lemma 4.** *Choose some node $u \in \mathcal{V}$. With probability at least $1 - 2\mathbf{K}_u^2 \exp\left(-\frac{n\epsilon_2^2}{2}\right)$, we have*

$$|\hat{H}_{k_1 k_2} - H_{k_1 k_2}| < \epsilon_2, \tag{59}$$

*for all $k_1, k_2 \in \mathcal{K}_u$.*

*Proof of Lemma 4.* Fix $k_1, k_2 \in \mathcal{K}_u$. Then the random variable defined as $Y_{k_1 k_2} = g_{uk_1}(\underline{\sigma}_{k_1}) g_{uk_2}(\underline{\sigma}_{k_2})$ satisfies $|Y_{k_1 k_2}| \leq 1$. Using the Hoeffding inequality we get

$$\mathbb{P}\left(|\hat{H}_{k_1 k_2} - H_{k_1 k_2}| > \epsilon_2\right) < 2\exp\left(-\frac{n\epsilon_2^2}{2}\right). \tag{60}$$

The proof follows by using the union bound over $k_1, k_2 \in \mathcal{K}_u$. $\qquad\square$

**Lemma 5.** *The residual of the first order Taylor expansion of the* GISO *satisfies*

$$\delta \mathcal{S}_n(\Delta, \underline{\theta}_u^*) \geq \exp(-\gamma) \frac{\Delta^T \hat{H} \Delta}{2 + \|\Delta\|_1}. \tag{61}$$

*Proof of Lemma 5.* Using Lemma 3 we have

$$\delta \mathcal{S}_n(\Delta, \theta_u^*) = \frac{1}{n} \sum_{t=1}^n \exp\left(-\sum_{k \in \mathcal{K}_u} \theta_k^* g_{uk}(\underline{\sigma}_k^{(t)})\right) \times \tag{62}$$

$$\left(\exp\left(-\sum_{k \in \mathcal{K}_u} \Delta_k g_{uk}(\underline{\sigma}_k^{(t)})\right) - 1 + \sum_{k \in \mathcal{K}_u} \Delta_k g_{uk}(\underline{\sigma}_k^{(t)})\right) \tag{63}$$

$$\geq \exp(-\gamma) \frac{\Delta^T \hat{H} \Delta}{2 + |\sum_{k \in \mathcal{K}_u} \Delta_k g_{uk}(\underline{\sigma}_k^{(t)})|}. \tag{64}$$

The proof follows by observing that $|\sum_{k \in \mathcal{K}_u} \Delta_k g_{uk}(\underline{\sigma}_k^{(t)})| \leq \|\Delta\|_1$. $\qquad\square$

We are now in a position to complete the proof of Proposition 4.

*Proof of Proposition 4: Restricted Strong Convexity for* GRISE. Using Lemma 5 we have

$$\delta \mathcal{S}_n(\Delta, \underline{\theta}_u^*) \geq \exp(-\gamma) \frac{\Delta^T \hat{H} \Delta}{2 + \|\Delta\|_1} \tag{65}$$

$$= \exp(-\gamma) \frac{\Delta^T H \Delta + \Delta^T (\hat{H} - H) \Delta}{2 + \|\Delta\|_1} \tag{66}$$

$$\overset{(a)}{\geq} \exp(-\gamma) \frac{\Delta^T H \Delta - \epsilon_2 \|\Delta\|_1^2}{2 + \|\Delta\|_1} \tag{67}$$

$$\overset{(b)}{\geq} \exp(-\gamma) \frac{\rho_u \|\Delta_{\mathcal{T}_u}\|^2 - \epsilon_2 \|\Delta\|_1^2}{2 + \|\Delta\|_1}. \tag{68}$$

where $(a)$ follows from Lemma 4 and $(b)$ follows from Condition 1 as

$$\Delta^T H \Delta = \mathbb{E}\left[\left(\sum_{k \in K_u} \Delta_k g_{uk}(\underline{\sigma}_k)\right)^2\right]. \tag{69}$$

$\qquad\square$

## C   Efficient implementation of GRISE and its computational complexity

The iterative Algorithm 2 takes as input a number of steps $T$ and output an $\epsilon$-optimal solution of GRISE without constraints in Eq. (74). This algorithm is an application of the Entropic Descent Algorithm introduced by [1] to reformulation of Eq. (13) as a minimization over the probability simplex. Note that there exist other efficient iterative methods for minimizing the GISO, such as the mirror gradient descent of [2]. The following proposition provides guarantees on the computational complexity of unconstrained GRISE.

**Proposition 5** (Computational Complexity for Unconstrained GRISE). *Let* $1 \geq \epsilon > 0$ *be the optimality gap and* $T \geq 6\epsilon^{-2} \ln(2\mathbf{K}_u + 1)$ *be the maximum number of iterations. Then Algorithm 2 is guaranteed to produce an $\epsilon$-optimal solution of* GRISE *without a constraint set* $\mathcal{Y}_u$ *with a number of operation less than* $C \frac{c_g n \mathbf{K}_u}{\epsilon^2} \ln(1 + \mathbf{K}_u)$, *where* $c_g$ *is an upper bound on the computational complexity of evaluating any* $g_{ik}(\underline{\sigma}_k)$ *for* $k \in \mathcal{K}_i$ *and* $C$ *is a universal constant that is independent of all parameters of the problem.*

**Algorithm 2:** Entropic Descent for unconstrained GRISE

```
// Step 1:  Initialization
```
1   $x^1_{k,+} \leftarrow 1/(2\mathbf{K}_u + 1), x^1_{k,-} \leftarrow 1/(2\mathbf{K}_u + 1), \forall k \in \mathcal{K}_u$;

2   $y^1 \leftarrow 1/(2\mathbf{K}_u + 1), \eta^1 \leftarrow \sqrt{2\ln(2\mathbf{K}_u + 1)}$;

```
// Step 2:  Entropic Descent Steps
```
3   **for** $t = 1, \ldots, T$ **do**

     `// Gradient Update:`

4      $v_k = \frac{\partial}{\partial \theta_k} \mathcal{S}(\widehat{\gamma}(\underline{x}^t_+ - \underline{x}^t_-)) / \mathcal{S}(\widehat{\gamma}(\underline{x}^t_+ - \underline{x}^t_-))$;

5      $w^+_k = x^t_{k,+}\exp(-\eta^t v_k), w^-_k = x^t_{k,-}\exp(\eta^t v_k)$.;

     `// Projection Step:`

6      $z = y^t + \sum_{k \in \mathcal{K}_u}(w^+_k + w^-_k)$;

7      $x^{t+1}_{k,+} \leftarrow \frac{w^+_k}{z}, x^{t+1}_{k,-} \leftarrow \frac{w^-_k}{z}$;

8      $y^{t+1} \leftarrow \frac{y^t}{z}$;

     `// Step Size Update:`

9      $\eta^{t+1} \leftarrow \eta^t \sqrt{\frac{t}{t+1}}$;

10   **end**

```
// Step 3:
```
11   $s = \operatorname{argmin}_{t=1,\ldots,T} \mathcal{S}(\widehat{\gamma}(\underline{x}^t_+ - \underline{x}^t_-))$;

12   **return** $\widehat{\underline{\theta}}_u = \widehat{\gamma}(\underline{x}^s_+ - \underline{x}^s_-)$;

---

*Proof of Proposition 5: Computational complexity of unconstrained* GRISE. We start by showing that the minimization of GRISE in Eq. (13) in the unconstrained case where $\mathcal{Y}_u = \mathbb{R}^{\mathbf{K}_u}$ is equivalent to the following lifted minimization on the logarithm of GRISE,

$$\min_{\underline{\theta}_u, \underline{x}^+, \underline{x}^-, y} \quad \log \mathcal{S}_n(\underline{\theta}_u) \tag{70}$$

$$\text{s.t.} \quad \underline{\theta}_u = \widehat{\gamma}(\underline{x}^+ - \underline{x}^-) \tag{71}$$

$$y + \sum_k (x^+_k + x^-_k) = 1 \tag{72}$$

$$y \geq 0, x^+_k \geq 0, x^-_k \geq 0, \forall k \in K_u. \tag{73}$$

We first show that for all $\underline{\theta}_u \in \mathbb{R}^{\mathbf{K}_u}$ such that $\|\underline{\theta}_u\|_1 \leq \widehat{\gamma}$, there exists $\underline{x}^+, \underline{x}^-, y$ satisfying constraints (71), (72), (73). This is easily done by choosing $x^+_k = \max(\theta_k/\widehat{\gamma}, 0)$, $x^-_k = \max(-\theta_k/\widehat{\gamma}, 0)$ and $y = 1 - \|\underline{\theta}_u\|_1/\widehat{\gamma}$. Second, we trivially see that for all $\underline{\theta}_u, \underline{x}^+, \underline{x}^-, y$ satisfying constraints (71), (72), (73), it implies that $\underline{\theta}_u$ also satisfies $\|\underline{\theta}_u\|_1 \leq \widehat{\gamma}$. Therefore, any $\underline{\theta}^{\min}_u$ that is an argmin of Eq. (70) is also an argmin of Eq. (13) without constraint set $\mathcal{Y}_u$. Moreover, if we find $\underline{\theta}^\epsilon_u$ such that $\log \mathcal{S}_n(\underline{\theta}^\epsilon_u) - \log \mathcal{S}_n(\underline{\theta}^{\min}_u) \leq \epsilon/\sqrt{3}$, we obtain an $\epsilon$-minimizer of Eq. (13) without constraint set $\mathcal{Y}_u$. Indeed, since $\epsilon/\sqrt{3} \leq \log(1 + \epsilon)$ for $1 \geq \epsilon > 0$, we have that $\mathcal{S}_n(\underline{\theta}^\epsilon_u) - \mathcal{S}_n(\underline{\theta}^{\min}_u) \leq \mathcal{S}_n(\underline{\theta}^{\min\,\epsilon}_u)\epsilon \leq \epsilon$ as $\mathcal{S}_n(\underline{\theta}^{\min}_u) \leq \mathcal{S}_n(0) = 1$. The remainder of the proof is a straightforward application of the analysis of the Entropic Descent Algorithm in [1, Th. 5.1] to the above minimization where $\underline{\theta}_u$ has been replaced by $\underline{x}^+, \underline{x}^-, y$ using Eq. (71). In this analysis we use the fact that the logarithm of GRISE remains a convex function as it is a sum of exponential functions and also that the gradient of our objective function is bounded uniformly by $\|\nabla \log \mathcal{S}_n(\underline{\theta}_u)\|_\infty = \|\nabla \mathcal{S}_n(\underline{\theta}_u)/\mathcal{S}_n(\underline{\theta}_u)\|_\infty \leq 1$ as $|g_k(\underline{\sigma}_k)| \leq 1$. Note that the computational complexity of the gradient evaluation is proportional to $n\mathbf{K}_u c_g$. This is because for each sample, one has to first compute an exponential containing $\mathbf{K}_u$ terms $g_k(\underline{\sigma}_k)$ with an evaluation cost of $c_g$ and then multiply the exponential by the factor $-g_k(\underline{\sigma}_k)$ corresponding to each of the $\mathbf{K}_u$ components of the gradient.   □

When the constraint set $\mathcal{Y}_u$ is parametrically complete, an $\epsilon$-optimal solution to (13) can be found by first solving the unconstrained version of GRISE and then performing an equi-cost projection onto

$\mathcal{Y}_u$. We define an $\epsilon$-optimal solution to the unconstrained GRISE problem as

$$\mathcal{S}_n(\widehat{\underline{\theta}}_u^{unc}) \leq \min_{\underline{\theta}_u : \|\underline{\theta}_u\|_1 \leq \widehat{\gamma}} \mathcal{S}_n(\underline{\theta}_u) + \epsilon. \tag{74}$$

**Lemma 6.** *Let $\widehat{\underline{\theta}}_u^{unc}$ be an $\epsilon$-optimal solution of the unconstrained GRISE problem in Eq. (74). Then an equi-cost projection of $\widehat{\underline{\theta}}_u^{unc}$ is an $\epsilon$-optimal solution of the constrained GRISE problem,*

$$\mathcal{S}_n(\mathcal{P}_{\mathcal{Y}_u}(\widehat{\underline{\theta}}_u^{unc})) \leq \min_{\underline{\theta}_u \in \mathcal{Y}_u : \|\underline{\theta}_u\|_1 \leq \widehat{\gamma}} \mathcal{S}_n(\underline{\theta}_u) + \epsilon. \tag{75}$$

*Proof of Lemma 6.* Since unconstrained GRISE is a relaxation of GRISE with the constraints $\underline{\theta}_u \in \mathcal{Y}_u$, we must have

$$\mathcal{S}_n(\widehat{\underline{\theta}}_u^{\mathrm{unc}}) \leq \min_{\underline{\theta}_u : \|\underline{\theta}_u\|_1 \leq \widehat{\gamma}} \mathcal{S}_n(\underline{\theta}_u) + \epsilon \leq \min_{\underline{\theta}_u \in \mathcal{Y}_u : \|\underline{\theta}_u\|_1 \leq \widehat{\gamma}} \mathcal{S}_n(\underline{\theta}_u) + \epsilon. \tag{76}$$

Since, $\mathcal{Y}_u$ is parametrically complete, by definition,

$$\mathcal{S}_n \left( \mathcal{P}_{\mathcal{Y}_u}(\widehat{\underline{\theta}}_u^{\mathrm{unc}}) \right) = \mathcal{S}_n(\widehat{\underline{\theta}}_u^{\mathrm{unc}}). \tag{77}$$

The estimates $\mathcal{P}_{\mathcal{Y}_u}(\widehat{\underline{\theta}}_u^{\mathrm{unc}})$ are feasible for the constrained GRISE problem, completing the proof. $\quad\square$

Lemma 6 implies that the computational complexity of GRISE for parametrically complete cases is the sum of the computational complexity of the unconstrained GRISE and the projection step.

---

**Algorithm 3:** Computing GRISE estimates for parametrically complete constraints

---

`// Step 1:  Solve unconstrained GRISE`
1 Use Algorithm 2 to obtain solutions $\widehat{\underline{\theta}}_u^{\mathrm{unc}}$ to the unconstrained GRISE ;
`// Step 2:  Perform projection step`
2 Project $\widehat{\underline{\theta}}_u^{\mathrm{unc}}$ onto $\mathcal{Y}_u$ to obtain the final estimates;
3 $\widehat{\underline{\theta}}_u = \mathcal{P}_{\mathcal{Y}_u}(\widehat{\underline{\theta}}_u^{\mathrm{unc}})$;
4 **return** $\widehat{\underline{\theta}}_u$;

---

Algorithm 3 is an implementation of GRISE for parametrically complete cases. Its computational complexity is obtained easily by combining Lemma 6 and Proposition 5.

**Theorem 3** (Computational Complexity for GRISE with P.C. Constraints). *Let $\mathcal{Y}_u$ be a parametrically complete set and let $1 \geq \epsilon > 0$ be given. Then Algorithm 3 computes an $\epsilon$-optimal solution to GRISE with a number of operations bounded by $C \frac{c_g n \mathbf{K}_u}{\epsilon^2} \ln(1 + \mathbf{K}_u) + \mathcal{C}(\mathcal{P}_{\mathcal{Y}_u}(\widehat{\underline{\theta}}_u^{unc}))$, where $c_g$ is an upper bound on the computational complexity of evaluating any $g_{ik}(\underline{\sigma}_k)$ for $k \in \mathcal{K}_i$ and where $\mathcal{C}(\mathcal{P}_{\mathcal{Y}_u}(\widehat{\underline{\theta}}_u^{unc}))$ denotes the computational complexity of the projection step.*

## D  Proofs & algorithms for structure and parameter estimation

### D.1  Dimension independence and easier computation of NPC constants

We recall the definition of the NPC constant,

$$\rho_i^{\mathrm{NPC}} = \min_{\substack{c \in \mathcal{M}_{\mathrm{cli}}(\mathcal{G}) \\ c \ni i}} \min_{\substack{\|\underline{x}_c\|_2 = 1 \\ \underline{x}_c \in \mathcal{X}_i^c}} \mathbb{E}_{(\sigma_i)} \left[ \sum_{\underline{\sigma}_{c \setminus i} \in \mathcal{A}_{c \setminus i}} \left( \sum_{k \in [c]_{\mathrm{sp}}} x_k h_k(\underline{\sigma}_k) \right)^2 \right]. \tag{78}$$

In order to give an intuition for the intricate formula in Eq. (78), let us define for maximal cliques $c \in \mathcal{M}_{\mathrm{cli}}(\mathcal{G})$, their clique parameterization matrix $G^c$, with indices being maximal factors $k, k' \in [c]_{\mathrm{sp}}$.

The clique parameterization matrix is obtained by summing over variable $\underline{\sigma}_c$ the globally centered basis functions,

$$G^c_{k,k'} = \sum_{\underline{\sigma}_c \in \mathcal{A}_c} h_k(\underline{\sigma}_c) h_{k'}(\underline{\sigma}_c). \tag{79}$$

Note that the clique parametertization matrices are positive semi-definite matrices by construction. Bounding the expectation over $\sigma_i$ using Lemma 1, we see that the NPC constant is linked to the smallest eigenvalue of the clique parameterization matrix,

$$\rho^{\mathrm{NPC}}_i \geq \frac{\exp(-2\gamma)}{q_i} \lambda_{\min}(G^c). \tag{80}$$

Clique parametertization matrices have a typical size of $\mathcal{O}(q^L \times q^L)$ since variables in a clique can take up to $\mathcal{O}(q^L)$ different configurations. Therefore, Eq. (80) emphasizes that that the NPC constant does not depend on the dimension of the model $p$ but rather on local properties of the parameterization of the family.

## D.2 Proofs for local learnability condition from nonsingular parametrization of cliques

*Proof of Proposition 1: LLC in $\ell_{\infty,2}$-norm.* For a given vertex $i \in \mathcal{V}$, let $\underline{x} \in \mathcal{X}_i \subseteq \mathbb{R}^{\mathbf{K}_i}$ be a vector in the perturbation set. First, suppose that $\{i\}$ is not a maximal clique and choose any maximal clique $c \in \mathcal{M}_{\mathrm{cli}}(\mathcal{G})$ that contains the vertex $i$ and let the set $S = c \setminus \{i\}$ be the set of nodes in the clique without $i$. The expression characterizing the LLC can be evaluated conditioning the expectation over $S$. Denoting the marginal and the conditional probability distribution used to compute expectation by subscripts, we find,

$$\mathbb{E}\left[\left(\sum_{k \in \mathcal{K}_i} x_k g_{ik}(\underline{\sigma}_k)\right)^2\right] = \mathbb{E}_{(\underline{\sigma}_{\backslash S})}\left[\mathbb{E}_{(\underline{\sigma}_S | \underline{\sigma}_{\backslash S})}\left[\left(\sum_{k \in \mathcal{K}_i} x_k g_{ik}(\underline{\sigma}_k)\right)^2\right]\right], \tag{81}$$

$$\geq \mathbb{E}_{(\underline{\sigma}_{\backslash S})}\left[\prod_{j \in S} \frac{\exp(-2\gamma)}{|\mathcal{A}_j|} \sum_{\underline{\sigma}_S \in \mathcal{A}_S}\left(\sum_{k \in \mathcal{K}_i} x_k g_{ik}(\underline{\sigma}_k)\right)^2\right], \tag{82}$$

where in the last line, we bounded the probability of $\mathbb{P}(\underline{\sigma}_S \mid \underline{\sigma}_{\backslash S})$ using Lemma 1. We want to rewrite the sum over $\underline{\sigma}_S$ in Eq. (82) using globally centered functions $h_k$ for factors $k \in [c]_{\mathrm{sp}}$ instead of locally centered functions $g_{ik}$. Using definitions of locally centered functions in Eq. (3) and globally centered functions in Eq. (22), we see that $g_{ik}(\underline{\sigma}_k) = h_k(\underline{\sigma}_k) + R_{ik}(\underline{\sigma}_k)$, where

$$R_{ik}(\underline{\sigma}_k) = -\sum_{\substack{r \in P(\partial k) \setminus \emptyset \\ r \neq \{i\}}} \frac{(-1)^{|r|}}{|\mathcal{A}_r|} \sum_{\underline{\sigma}_r} f_k(\underline{\sigma}_k). \tag{83}$$

The sum in Eq. (82) can be expanded into the four contributions,

$$\sum_{\underline{\sigma}_S \in \mathcal{A}_S}\left(\sum_{k \in \mathcal{K}_i} x_k g_{ik}(\underline{\sigma}_k)\right)^2 = \sum_{\underline{\sigma}_S \in \mathcal{A}_S}\left(\sum_{k \in [c]_{\mathrm{sp}}} x_k h_k(\underline{\sigma}_c)\right)^2 \tag{84}$$

$$+ \sum_{k \in [c]_{\mathrm{sp}}} \sum_{l \in \mathcal{K}_i \setminus [c]_{\mathrm{sp}}} x_k x_l \sum_{\underline{\sigma}_S \in \mathcal{A}_S} h_k(\underline{\sigma}_c) g_{il}(\underline{\sigma}_l) \tag{85}$$

$$+ \sum_{k \in [c]_{\mathrm{sp}}} \sum_{l \in [c]_{\mathrm{sp}}} x_k x_l \sum_{\underline{\sigma}_S \in \mathcal{A}_S} h_k(\underline{\sigma}_c) R_{il}(\underline{\sigma}_c) \tag{86}$$

$$+ \sum_{\underline{\sigma}_S \in \mathcal{A}_S}\left(\sum_{k \in \mathcal{K}_i \setminus [c]_{\mathrm{sp}}} x_k g_{ik}(\underline{\sigma}_k) + \sum_{k \in [c]_{\mathrm{sp}}} x_k R_{ik}(\underline{\sigma}_c)\right)^2. \tag{87}$$

We start by evaluating the contribution from terms in Eq. (85). For $k \in [c]_{\mathrm{sp}}$ and $l \in \mathcal{K}_i \setminus [c]_{\mathrm{sp}}$, there exists at least one node $u \in c$ such that $u \neq i$ and $u \notin \partial l$. Summing over the variable $\sigma_u$ cancels the

expression,

$$\sum_{\sigma_u \in \mathcal{A}_u} h_k(\underline{\sigma}_c) g_{il}(\underline{\sigma}_l) = 0, \tag{88}$$

as $h_k(\underline{\sigma}_c)$ is globally centered and $g_{il}(\underline{\sigma}_l)$ does not depend on $\sigma_u$.

The contribution from Eq. (86) is also null. To see this we expand the sums using the formula for the reminder in Eq. (83),

$$\sum_{\underline{\sigma}_S \in \mathcal{A}_S} h_k(\underline{\sigma}_c) R_{il}(\underline{\sigma}_c) = -\sum_{\substack{r \in P(\partial l)\setminus\emptyset \\ r \neq \{i\}}} \frac{(-1)^{|r|}}{|\mathcal{A}_r|} \sum_{\underline{\sigma}_S \in \mathcal{A}_S} h_k(\underline{\sigma}_c) \sum_{\underline{\sigma}_r} f_l(\underline{\sigma}_c) = 0, \tag{89}$$

where the sum over $\underline{\sigma}_S$ vanishes as $h_k(\underline{\sigma}_c)$ depends on $\underline{\sigma}_r \neq \sigma_i$ while $\sum_{\underline{\sigma}_r} f_l(\underline{\sigma}_c)$ does not. As the contribution from Eq.(87) is non-negative, we can lower-bound Eq. (82) by the following expression,

$$\mathbb{E}\left[\left(\sum_{k \in \mathcal{K}_i} x_k g_{ik}(\underline{\sigma}_k)\right)^2\right] \geq \prod_{j \in S} \frac{\exp(-2\gamma)}{|\mathcal{A}_j|} \mathbb{E}_{(\underline{\sigma}_{\setminus S})}\left[\sum_{\underline{\sigma}_S \in \mathcal{A}_S} \left(\sum_{k \in [c]_{\mathrm{sp}}} x_k h_k(\underline{\sigma}_c)\right)^2\right], \tag{90}$$

$$\geq \prod_{j \in S} \frac{\exp(-2\gamma)}{|\mathcal{A}_j|} \rho_i^{\mathrm{NPC}} \sum_{k \in [c]_{\mathrm{sp}}} x_k^2, \tag{91}$$

where in the last line we have recognized the definition of the NPC constant from Eq. (23). Since (91) holds for any $c \in \mathcal{M}_{\mathrm{cli}}(\mathcal{G})$ that contains the vertex $i$, the Local Learnability Condition is satisfied for a weighted $\ell_{\infty,2}$-norm with LLC constant equal to $\rho^{\mathrm{NPC}}$,

$$\mathbb{E}\left[\left(\sum_{k \in \mathcal{K}_i} x_k g_{ik}(\underline{\sigma}_k)\right)^2\right] \geq \rho_i^{\mathrm{NPC}} \|\underline{x}_{\mathcal{T}_i}\|_{w(\infty,2)}^2, \tag{92}$$

where the weighted $\ell_{\infty,2}$-norm is defined as follows,

$$\|\underline{x}_{\mathcal{T}_i}\|_{w(\infty,2)} = \max_{\substack{c \ni i \\ c \in \mathcal{M}_{\mathrm{cli}}(\mathcal{G})}} \sqrt{\prod_{j \in c\setminus\{i\}} \frac{\exp(-2\gamma)}{|\mathcal{A}_j|} \sum_{k \in [c]_{\mathrm{sp}}} x_k^2}. \tag{93}$$

As the weighted $\ell_{\infty,2}$-norm in Eq. (93) is lower-bounded by the $\ell_{\infty,2}$-norm,

$$\|\underline{x}_{\mathcal{T}_i}\|_{w(\infty,2)}^2 \geq \left(\frac{\exp(-2\gamma)}{q}\right)^{L-1} \|\underline{x}_{\mathcal{T}_i}\|_{\infty,2}^2, \tag{94}$$

we have that the LLC is also satisfied for the $\ell_{\infty,2}$-norm with LLC constant equal to $\rho_i^{\mathrm{NPC}}\left(\frac{\exp(-2\gamma)}{q}\right)^{L-1}$.

When $\{i\}$ is a maximal clique, then $\mathcal{K}_i = [\{i\}]_{\mathrm{sp}}$ and it straightforward to see that

$$\mathbb{E}\left[\left(\sum_{k \in \mathcal{K}_i} x_k g_{ik}(\underline{\sigma}_k)\right)^2\right] = \mathbb{E}_{(\sigma_i)}\left[\left(\sum_{k \in [\{i\}]_{\mathrm{sp}}} \underline{x}_k h_k(\sigma_i)\right)^2\right] \tag{95}$$

$$\geq \rho_i^{\mathrm{NPC}} \sum_{k \in [\{i\}]_{\mathrm{sp}}} x_k^2. \tag{96}$$

$\square$

**Lemma 7.** *Let $\sigma \in \mathcal{A}$, be a discrete random variable with probability distribution $p(\sigma)$. Consider $x_\sigma \in \mathbb{R}$, a function defined over $\sigma$ that is centered, i.e. $\sum_{\sigma \in \mathcal{A}} x_\sigma = 0$. The variance of the function $x_\sigma$ is lower-bounded by,*

$$\mathrm{Var}\left[x_\sigma\right] \geq p_{min} \sum_{\sigma \in \mathcal{A}} x_\sigma^2, \tag{97}$$

*where $p_{min} = \min_{\sigma \in \mathcal{A}} p(\sigma)$.*

*Proof.* The proof goes as follows,

$$\text{Var}\,[x_\sigma] = \sum_{\sigma \in \mathcal{A}} p(\sigma) \left( x_\sigma - \sum_{\sigma' \in \mathcal{A}} p(\sigma') x_{\sigma'} \right)^2 \tag{98}$$

$$\geq p_{\min} \sum_{\sigma \in \mathcal{A}} \left( x_\sigma - \sum_{\sigma' \in \mathcal{A}} p(\sigma') x_{\sigma'} \right)^2 \tag{99}$$

$$= p_{\min} \sum_{\sigma \in \mathcal{A}} \left( x_\sigma^2 - 2 x_\sigma \sum_{\sigma' \in \mathcal{A}} p(\sigma') x_{\sigma'} + \left( \sum_{\sigma' \in \mathcal{A}} p(\sigma') x_{\sigma'} \right)^2 \right), \tag{100}$$

$$\geq p_{\min} \sum_{\sigma \in \mathcal{A}} x_\sigma^2, \tag{101}$$

where in the last line we used that $\sum_{\sigma \in \mathcal{A}} x_\sigma = 0$ and $\left( \sum_{\sigma' \in \mathcal{A}} p(\sigma') x_{\sigma'} \right)^2 \geq 0$. $\qquad\square$

*Proof of Proposition 2: LLC in $\ell_2$-norm for pairwise models.* For a given vertex $i \in \mathcal{V}$, let $\underline{x} \in \mathcal{X}_i \subseteq \mathbb{R}^{\mathbf{K}_i}$ be a vector in the perturbation set. When $\{i\}$ is a maximal clique, then $\mathcal{K}_i = [\{i\}]_{\text{sp}}$ and we immediately see that

$$\mathbb{E}\left[ \left( \sum_{k \in \mathcal{K}_i} x_k g_{ik}(\underline{\sigma}_k) \right)^2 \right] = \mathbb{E}_{(\sigma_i)} \left[ \left( \sum_{k \in [\{i\}]_{\text{sp}}} \underline{x}_k h_k(\sigma_i) \right)^2 \right] \tag{102}$$

$$\geq \rho_i^{\text{NPC}} \sum_{k \in [\{i\}]_{\text{sp}}} x_k^2. \tag{103}$$

Now suppose that $\{i\}$ is not a maximal clique, i.e. there exists $j \in \mathcal{V}$ such that $\{i, j\} \in \mathcal{M}_{\text{cli}}\,(\mathcal{G}^*)$. The expectation that arises in the LLC is lower-bounded by its variance,

$$\mathbb{E}\left[ \left( \sum_{k \in \mathcal{K}_i} x_k g_{ik}(\underline{\sigma}_k) \right)^2 \right] \geq \text{Var}\left[ \sum_{k \in \mathcal{K}_i} x_k g_{ik}(\underline{\sigma}_k) \right]. \tag{104}$$

Let $\{S_r\}_{r=1,\ldots,\chi}$ be a minimal coloring of the graph $\mathcal{G}^*$. For a given color $r$, define the set $C_r = S_r \setminus \{i\}$ and apply the law of total variance on the right-hand side of Eq. (104), conditioning on $\underline{\sigma}_{\setminus C_r}$,

$$\text{Var}\left[ \sum_{k \in \mathcal{K}_i} x_k g_{ik}(\underline{\sigma}_k) \right] \geq \mathbb{E}_{(\underline{\sigma}_{\setminus C_r})} \left[ \text{Var}_{(\underline{\sigma}_{C_r} | \underline{\sigma}_{\setminus C_r})} \left[ \sum_{k \in \mathcal{K}_i} x_k g_{ik}(\underline{\sigma}_k) \right] \right], \tag{105}$$

where the marginal and the conditional probability distribution used to compute expectation and variance respectively are indicated by subscripts. As the variance on the right-hand side of Eq. (105) is conditioned on $\underline{\sigma}_{\setminus C_r}$, only basis functions involving a pair $(\sigma_i, \sigma_j)$ with $j \in C_r$ are giving a non-zero contribution to the conditional variance,

$$\text{Var}_{(\underline{\sigma}_{C_r} | \underline{\sigma}_{\setminus C_r})} \left[ \sum_{k \in \mathcal{K}_i} x_k g_{ik}(\underline{\sigma}_k) \right] = \text{Var}_{(\underline{\sigma}_{C_r} | \underline{\sigma}_{\setminus C_r})} \left[ \sum_{j \in C_r} \sum_{k \in [\{i,j\}]_{\text{sp}}} x_k g_{ik}(\sigma_i, \sigma_j) \right]. \tag{106}$$

We can rewrite the locally centered functions with respect to globally centered functions using their definitions found in Eq. (3) and in Eq. (22),

$$g_{ik}(\sigma_i, \sigma_j) = h_k(\sigma_i, \sigma_j) - \frac{1}{|\mathcal{A}_i|} \sum_{\sigma_j} f_k(\sigma_i, \sigma_j) + \frac{1}{|\mathcal{A}_i||\mathcal{A}_j|} \sum_{\sigma_i, \sigma_j} f_k(\sigma_i, \sigma_j). \tag{107}$$

We see from Eq. (107) that the difference between locally and globally centered functions only depends on the variable $\sigma_i$. This means that we can interchange locally centered functions with

globally centered functions in the right-hand side of Eq. (106) as the variance is conditioned on $\sigma_i$,

$$\text{Var}_{(\underline{\sigma}_{C_r}|\underline{\sigma}_{\backslash C_r})}\left[\sum_{j\in C_r}\sum_{k\in[\{i,j\}]_{\text{sp}}} x_k g_{ik}(\sigma_i,\sigma_j)\right] = \text{Var}_{(\underline{\sigma}_{C_r}|\underline{\sigma}_{\backslash C_r})}\left[\sum_{j\in C_r}\sum_{k\in[\{i,j\}]_{\text{sp}}} x_k h_k(\sigma_i,\sigma_j)\right]. \tag{108}$$

Since $\{S_r\}_{r=1,\dots,\chi}$ is a vertex coloring, by definition all nodes $j\in C_r$ having the same color are not sharing a factor node, i.e. $\forall j_1,j_2\in C_r$, $\nexists k\in\mathcal{K}^*$ such that $j_1,j_2\in\partial k$. This implies that variables $\sigma_j$ with $j\in C_r$ are independent conditioned on the remaining variables $\underline{\sigma}_{\backslash C_r}$ and the variance in Eq. (108) can be rewritten,

$$\text{Var}_{(\underline{\sigma}_{C_r}|\underline{\sigma}_{\backslash C_r})}\left[\sum_{j\in C_r}\sum_{k\in[\{i,j\}]_{\text{sp}}} x_k h_k(\sigma_i,\sigma_j)\right] = \sum_{j\in C_r}\text{Var}_{(\underline{\sigma}_j|\underline{\sigma}_{\backslash C_r})}\left[\sum_{k\in[\{i,j\}]_{\text{sp}}} x_k h_k(\sigma_i,\sigma_j)\right]. \tag{109}$$

The right-hand side of Eq. (109) is centered with respect to $\sigma_j$ and we can apply Lemma 7 and Lemma 1 to find a lower-bound that is only dependant on the random variable $\sigma_i$,

$$\sum_{j\in C_r}\text{Var}_{(\underline{\sigma}_j|\underline{\sigma}_{\backslash C_r})}\left[\sum_{k\in[\{i,j\}]_{\text{sp}}} x_k h_k(\sigma_i,\sigma_j)\right] \geq \frac{\exp(-2\gamma)}{q}\sum_{j\in C_r}\sum_{\sigma_j\in\mathcal{A}_j}\left(\sum_{k\in[\{i,j\}]_{\text{sp}}} x_k h_k(\sigma_i,\sigma_j)\right)^2. \tag{110}$$

Plugging back the results derived in Eq. (108), Eq. (109), and Eq. (110) into the initial inequality in Eq. (105), we find,

$$\text{Var}\left[\sum_{k\in\mathcal{K}_i} x_k g_{ik}(\underline{\sigma}_k)\right] \geq \mathbb{E}_{(\sigma_i)}\left[\frac{\exp(-2\gamma)}{q}\sum_{j\in C_r}\sum_{\sigma_j\in\mathcal{A}_j}\left(\sum_{k\in[\{i,j\}]_{\text{sp}}} x_k h_k(\sigma_i,\sigma_j)\right)^2\right], \tag{111}$$

$$\geq \frac{\exp(-2\gamma)}{q}\rho_i^{\text{NPC}}\sum_{j\in C_r}\sum_{k\in[\{i,j\}]_{\text{sp}}} x_k^2, \tag{112}$$

where in Eq. (112) we used the definition of the NPC constant in Eq. (23) to bound the quadratic form involving $\underline{x}$.

Finally, we average the inequality described by Eq. (112) over the different colors and hence possible conditioning sets $C_r$ to conclude the proof,

$$\text{Var}\left[\sum_{k\in\mathcal{K}_i} x_k g_{ik}(\underline{\sigma}_k)\right] \geq \frac{\exp(-2\gamma)}{q}\frac{\rho_i^{\text{NPC}}}{\chi}\sum_{r=1,\dots,\chi}\sum_{j\in C_r}\sum_{k\in[\{i,j\}]_{\text{sp}}} x_k^2, \tag{113}$$

$$= \frac{\exp(-2\gamma)}{q}\frac{\rho_i^{\text{NPC}}}{\chi}\sum_{\substack{k\in\mathcal{M}_{\text{fac}}(\mathcal{G})\\\partial k\ni i}} x_k^2. \tag{114}$$

□

### D.3 Proofs of estimation guarantees for the SUPRISE algorithm

*Proof of Theorem 2: Reconstruction and Estimation Guarantees for* SUPRISE. As the NPC constant is non-zero $\rho_u^{\text{NPC}} > 0$ for all nodes $u\in\mathcal{V}$, we apply Proposition 1 in conjunction with Theorem 1 to find that for each step $t\in\{0,\dots,L-1\}$ and with probability at least $1-\delta/(pL)$, GRISE around a node $u\in\mathcal{V}$ recovers the parameters in each maximal clique $c\in\mathcal{M}_{\text{cli}}\left(\mathcal{G}\left[(\mathcal{V},\mathcal{K}^t)\right]\right)$ that contains $u$ with precision $\sum_{k\in[c]_{\text{sp}}}(\theta_k-\widehat{\theta}_k)^2\leq(\alpha/2)^2$. Therefore, at each step $t\in\{0,\dots,L-1\}$ and with probability at least $1-\delta/L$, the factor removal procedure is guaranteed to remove all factors of size $L-t$ that are not present in the graph if all factors of size bigger than $L-t$ were correctly removed in the previous steps. Since there are at most $L$ removal steps, it implies that the overall procedure discovers all maximal cliques with probability at least $1-\delta$.

□

# E Application to special cases

In this section, we show how to apply Theorem 2 in order to derive the sample and computational complexity of reconstructing graphical models for some common basis functions.

## E.1 Binary models on the monomial basis

In this subsection, we consider general models on binary alphabet $\mathcal{A}_i = \{-1, 1\}$. Let the factors be all nonempty subsets of $\{1, \ldots, p\} = \mathcal{V}$ of size at most $L$,

$$\mathcal{K} = \{k \subseteq \mathcal{V} \mid |k| \leq L\}. \tag{115}$$

The set $\mathcal{K}$ contains all potential subsets of variable of size at most $L$. The parameterization uses the monomial basis given by $f_k(\underline{\sigma}_k) = \prod_{j \in k} \sigma_j$ with $k \in \mathcal{K}$. Note that the monomial basis functions are already globally centered $f_k \equiv g_k \equiv h_k$. The probability distribution for this model is expressed as

$$\mu_{\text{binary}}(\underline{\sigma}) = \frac{1}{Z} \exp \left( \sum_{k \in \mathcal{K}} \theta_k^* \prod_{j \in k} \sigma_j \right). \tag{116}$$

When $L \leq 2$, the model in Eq. (116) is pairwise and it is referred as the Ising Model.

For each maximal clique there exists exactly one maximal factors in its span. Therefore, the NPC constant as defined in Eq. (23) is $\rho_{\text{NPC}} = 1$ since for any clique $c$ we have,

$$\sum_{\underline{\sigma}_{c \setminus u} \in \mathcal{A}_{c \setminus u}} \left( \sum_{k \in [c]_{\text{sp}}} x_k h_k(\underline{\sigma}_k) \right)^2 = x_k^2 \sum_{\underline{\sigma} \in \{-1,1\}^{|c|-1}} \left( \prod_{i=1}^{|c|} \sigma_i \right)^2 = 2^{|c|-1} x_k^2, \tag{117}$$

and the minimum is achieved for cliques of size one. As every node is involved in at most $\mathbf{K} \leq p^{L-1}$ factor functions, the structure of binary models can be recovered as a corollary of Theorem 2.

**Corollary 1 (Structure recovery for binary graphical models).** *Let $\underline{\sigma}^{(1)}, \ldots, \underline{\sigma}^{(n)}$ be i.i.d. samples drawn according to $\mu(\underline{\sigma})$ in (116) and let $\alpha \leq \min_{k \in \mathbb{S}(\mathcal{G}^*)} |\theta_k^*|$ be the intensity of the smallest non-zero parameter. If*

$$n \geq 2^{12} 4^L \frac{\widehat{\gamma}^2 (1+\widehat{\gamma})^2 e^{4\gamma L}}{\alpha^4} \log \left( \frac{4Lp^{2L-1}}{\delta} \right), \tag{118}$$

*then the structure of the binary graphical model is perfectly recovered using Algorithm 1, i.e. $\widehat{\mathbb{S}} = \mathbb{S}(\underline{\theta}^*)$, with probability $1 - \delta$. Moreover the total computational complexity scales as $\widetilde{\mathcal{O}}(p^L)$, for fixed $L$, $\alpha$, $\gamma$, $\widehat{\gamma}$ and $\delta$.*

For pairwise Ising models that are $\chi$ colorable, we have also guarantees on the $\ell_2$-norm reconstruction by SUPRISE of pairwise parameters.

**Corollary 2 ($\ell_2$-parameter estimation for Ising models).** *Let $\underline{\sigma}^{(1)}, \ldots, \underline{\sigma}^{(n)}$ be i.i.d. samples drawn according to $\mu(\underline{\sigma})$ in (116) for $L = 2$ and let $\alpha > 0$ be the prescribed estimation accuracy. If*

$$n \geq 2^{16} \frac{\widehat{\gamma}^2 (1+\widehat{\gamma})^2 \chi^2 e^{8\gamma}}{\alpha^4} \log \left( \frac{8p^3}{\delta} \right), \tag{119}$$

*then, with probability at least $1 - \delta$, the parameters are estimated by Algorithm 1 with the error*

$$\sqrt{\sum_{i,j \in V} |\widehat{\underline{\theta}}_{ij} - \underline{\theta}_{ij}^*|^2} \leq \frac{\alpha}{2}. \tag{120}$$

*The computational complexity of obtaining these estimates is $\widetilde{\mathcal{O}}(p^2)$ for fixed $\chi$, $\alpha$, $\gamma$, $\widehat{\gamma}$ and $\delta$.*

As graphs with bounded degree $d$ have a chromatic number at most $d + 1 \geq \chi$, Corollary 2 recovers the $\ell_2$-guarantees for sparse graphs recovery of [18] albeit with slightly worse dependence with respect to $\gamma$ and $\alpha$. The worse $\gamma$ dependence is an artifact of the general analysis presented in this paper. For models over binary variables one can improve the $e^{8\gamma}$ dependence to $e^{6\gamma}$ using Berstein's

inequality in Proposition 3 instead of Hoeffding's inequality. However, the worse $\alpha$ dependence seems to be more fundamental. It is caused by the replacement of the $\ell_1$-penalty used in [18] by an $\ell_1$-constraint.

For graphs with unbounded vertex degree but low chromatic number, such as star graphs or bipartite graphs, Corollary 2 shows that the parameters of the corresponding Ising model can be fully recovered with a bounded $\ell_2$-error using a number of samples that is only logarithmic in the model size $p$.

### E.2  L-wise models with arbitrary alphabets on the indicator basis

In this subsection, we consider $L$-wise graphical models over variables taking values in arbitrary alphabet $\mathcal{A}_i$ of size $q_i$, parametrized with indicator-type functions. The building block of the set of basis functions is the centered univariate indicator function defined as

$$\Phi_{s_i,\sigma_i} = \begin{cases} 1 - \frac{1}{q_i}, & \text{if } \sigma_i = s_i, \\ -\frac{1}{q_i}, & \text{otherwise,} \end{cases} \tag{121}$$

where $s_i, \sigma_i \in \mathcal{A}_i$ are prescribed letters of the alphabet. The univariate indicator functions in Eq. (121) are centered Kronecker delta functions and possess similar properties such as symmetry $\Phi_{s_i,\sigma_i} = \Phi_{\sigma_i,s_i}$ and contraction under a summation,

$$\sum_{\tau_i \in \mathcal{A}_i} \Phi_{\tau_i,s_i} \Phi_{\tau_i,\sigma_i} = \Phi_{s_i,\sigma_i}. \tag{122}$$

The set of factors $\mathcal{K}$ are pairs associating elements of $R = \{r \in P(\mathcal{V}) \mid |r| \leq L\}$ which are subsets of variable of size at most $L$ with an alphabet configuration in $\mathcal{A}_r = \bigotimes_{i \in r} \mathcal{A}_i$,

$$\mathcal{K} = \{(r, \underline{s}_r) \mid r \in R, \underline{s}_r \in \mathcal{A}_r\}. \tag{123}$$

In what follows, we slightly abuse the notation of factors and parameters by shortening $(r, \underline{s}_r) \equiv \underline{s}_r$. With these notations, the indicator basis functions are constructed as $f_{\underline{s}_r}(\underline{\sigma}_r) = \prod_{i \in r} \Phi_{s_i,\sigma_i}$. Note that the indicator basis functions are globally centered i.e. $f_{\underline{s}_r} \equiv g_{\underline{s}_r} \equiv h_{\underline{s}_r}$. The probability distribution of an $L$-wise graphical model with arbitrary alphabet is defined as follows,

$$\mu_{\text{general}}(\underline{\sigma}) = \frac{1}{Z} \exp\left( \sum_{r \in R} \sum_{\underline{s}_r \in \mathcal{A}_r} \theta^*_{\underline{s}_r} \prod_{i \in r} \Phi_{s_i,\sigma_i} \right). \tag{124}$$

The family of distribution in Eq. (124) is not uniquely parameterized by the parameters $\underline{\theta}^*$. To see this, we introduce the linear application $\mathcal{P}_r$ acting on arrays $\theta_{\underline{s}_r}$ as follows,

$$[\mathcal{P}_r \underline{\theta}]_{\underline{\sigma}_r} = \sum_{\underline{s}_r \in \mathcal{A}_r} \theta_{\underline{s}_r} \prod_{i \in r} \Phi_{s_i,\sigma_i}. \tag{125}$$

Using the contraction property from Eq. (122), it is easy to see that $\mathcal{P}_r$ is a projector, i.e $\mathcal{P}_r^2 = \mathcal{P}_r$. It is also straightforward to verify that $\mathcal{P}_r \underline{\theta}$ is always a globally centered array and if $\underline{\theta}$ is already globally centered then $\mathcal{P}_r \underline{\theta} = \underline{\theta}$. Therefore, the applications $\mathcal{P}_r$ are projectors on the space of array $\theta_{\underline{s}_r}$ which are globally centered, i.e. $\sum_{s_i} \theta_{\underline{s}_r} = 0$ for all $i \in r$. We lift the parametrization degeneracy in Eq. (124) by imposing that parameters $\underline{\theta}^*$ are in the range of the projector $\mathcal{P}_r$. We thus require that the parameters satisfy the following linear constraints at each vertex $u \in \mathcal{V}$,

$$\mathcal{Y}_u = \bigcap_{\substack{r \in R \\ r \ni u}} \left\{ \underline{\theta}_u \in \mathbb{R}^{\mathbf{K}_u} \;\middle|\; \forall i \in r, \sum_{s_i \in \mathcal{A}_i} \theta_{\underline{s}_r} = 0 \right\}. \tag{126}$$

The constraint set in Eq. (126) is parametrically complete according to Definition 2 as we explicitly exhibited the equi-cost projection $\{\mathcal{P}_r\}_{r \in R}$ onto it. The computational complexity of this projection is no more than $\mathcal{O}(p^{L-1} q^L)$.

As the constraint set in Eq. (126) forms a linear subspace, the perturbation set is simply $\mathcal{X}_u = \mathcal{Y}_u \cap B_1(2\widehat{\gamma})$, the intersection of the constraint set with the $\ell_1$-ball of radius $2\widehat{\gamma}$. Maximal cliques

are subset of vertices and hence are also elements of $R$. Therefore, the NPC constant as defined in Eq. (23) is bounded by $\rho_{\text{NPC}} \geq \exp(-2\gamma)/q$ since for each clique we have,

$$\mathbb{E}_{(\sigma_i)}\left[\sum_{\underline{\sigma}_{c\backslash i}\in\mathcal{A}_{c\backslash i}}\left(\sum_{k\in[c]_{\text{sp}}}x_k h_k(\underline{\sigma}_k)\right)^2\right] \geq \frac{\exp(-2\gamma)}{q_i}\sum_{\underline{\sigma}_c\in\mathcal{A}_c}\left([\mathcal{P}_c\underline{x}]_{\underline{\sigma}_c}\right)^2, \qquad (127)$$

$$= \frac{\exp(-2\gamma)}{q_i}\sum_{\underline{\sigma}_c\in\mathcal{A}_c}x_{\underline{\sigma}_c}^2, \qquad (128)$$

as $\underline{x}\in\mathcal{X}_u$ is globally centered and thus is in the range of the projector $\mathcal{P}_c$. Every node is involved in at most $\mathbf{K}\leq p^{L-1}q^L$ factor functions and the structure of L-wise models with arbitrary alphabets can be recovered as a corollary of Theorem 2.

**Corollary 3 (Structure recovery for L-wise graphical models).** *Let $\underline{\sigma}^{(1)},\ldots,\underline{\sigma}^{(n)}$ be i.i.d. samples drawn according to $\mu(\underline{\sigma})$ in (124) and let $\alpha \leq \min_{c\in\mathbb{S}(\mathcal{G}^*)}\sqrt{\sum_{\underline{s}_c\in\mathcal{A}_c}\theta_{\underline{s}_c}^{*\ 2}}$ be the intensity of the smallest non-zero parameter. If*

$$n \geq 2^{14}q^{2L}\frac{\widehat{\gamma}^2(1+\widehat{\gamma})^2 e^{4\gamma(L+1)}}{\alpha^4}\log\left(\frac{4Lq^{2L}p^{2L-1}}{\delta}\right), \qquad (129)$$

*then the structure of the L-wise graphical model with arbitrary alphabets is perfectly recovered using Algorithm 1, i.e. $\widehat{\mathbb{S}} = \mathbb{S}(\underline{\theta}^*)$, with probability $1-\delta$. Moreover the total computational complexity scales as $\widetilde{\mathcal{O}}(p^L)$, for fixed $L$, $q$, $\alpha$, $\gamma$, $\widehat{\gamma}$ and $\delta$.*

For pairwise models with arbitrary alphabet that are $\chi$ colorable, we have also guarantees on the $\ell_2$-norm reconstruction by SUPRISE of pairwise parameters.

**Corollary 4 ($\ell_2$-parameter estimation for pairwise models).** *Let $\underline{\sigma}^{(1)},\ldots,\underline{\sigma}^{(n)}$ be i.i.d. samples drawn according to $\mu(\underline{\sigma})$ in (124) for $L=2$ and let $\alpha > 0$ be the prescribed estimation accuracy. If*

$$n \geq 2^{14}q^4\frac{\widehat{\gamma}^2(1+\widehat{\gamma})^2\chi^2 e^{12\gamma}}{\alpha^4}\log\left(\frac{8q^4p^3}{\delta}\right), \qquad (130)$$

*then, with probability at least $1-\delta$, the parameters are estimated by Algorithm 1 with the error*

$$\sqrt{\sum_{i,j\in V}\sum_{s_i\in\mathcal{A}_i,s_j\in\mathcal{A}_j}|\widehat{\underline{\theta}}_{s_i,s_j}-\underline{\theta}_{s_i,s_j}^*|^2} \leq \frac{\alpha}{2}. \qquad (131)$$

*The computational complexity of obtaining these estimates is $\widetilde{\mathcal{O}}(p^2)$ for fixed $\chi$, $q$, $\alpha$, $\gamma$, $\widehat{\gamma}$ and $\delta$.*