[Reviews · NeurIPS 2020]

Review 1

Summary and Contributions: Update after author rebuttal phase: I stand by my initial evaluation of the paper, and recommend acceptance. -------------- This paper studies the problem of learning discrete factor models from data. The authors propose a condition that allows them to provide guarantees on the model selection and parameter estimation. They also propose an algorithm for learning the factor graph based on interaction screening. Finally, the authors propose a specialization of this algorithm that can perform structure and parameter learning for graphical models.

Strengths: - The theoretical results are sound and interesting. This seems like the natural generalization of a line of work based on interaction screening for understanding model selection in discrete graphical models. - this is a significant problem of interest to the NeurIPS community - The conditions that the authors propose -- local learnability -- seems general and specializes in a way that seems "right"

Weaknesses: nothing significant

Correctness: The claims seem correct to the best of my reading

Clarity: The paper is generally well written. I have some minor comments: - The authors should include a more detailed discussion on Condition C1 in the main text of the paper given its centrality to the paper. - the authors claim that the proposed framework includes all models previously considered in the literature as special cases, and that their analysis shows a systematic improvement in sample complexity. These comparisons shouldn't be relegated to the appendix, and should be included in the main text -- preferably as a table.

Relation to Prior Work: The relation to prior work is clearly discussed -- it will help if some of this material is moved from the appendix to the main text.

Reproducibility: Yes

Additional Feedback:


Review 2

Summary and Contributions: The paper presents a general method for learning undirected discrete probabilistic graphical models with provable and somewhat tight upper bounds on the required number of independent samples and computation time. Specifically, the work generalizes the so-called interaction screening method, originally introduced for pairwise binary models. ADDED: Thanks for the rebuttal. I found the explanation in Supplement D.1 for the efficient verifiability of the identifiability condition hard to follow. I presume the needed values are given by Eq. (82). It is not at all clear to me why the formula in Eq. (82) is efficient to evaluate. (Eq. (84) only gives a lower bound and cannot, of course, not replace the exact formula.)

Strengths: + Extends and improves previous results. + A challenging research problem studied by several research groups.

Weaknesses: - The work is purely theoretical. It is not clear whether the presented algorithms could be feasible in practice. - The presentation is sometimes hard to follow. E.g., Lines 60-61 seem to promise that the indentifiability condition can be verified in an efficient way, but it was difficult to figure out what this way actually is.

Correctness: I could not spot any errors.

Clarity: There are minor problems in writing style. Otherwise the authors seem to have done good job in coordinating the reader through the material.

Relation to Prior Work: Prior research is well discussed, as far as I can tell.

Reproducibility: Yes

Additional Feedback: Minor comments: - There are some bad notational choices (in my opinion), e.g., K_i in bold and underlining for vectors. - "[p]" is not defined. - In Def 1, what is alpha and who specifies it? Why absolute error, not relative error? - I'd recommend the lazy citation style, e.g., "analyzed in [17]", by proper phrases, e.g., "analyzed by "Vuffray et al. [17]"; the sentence should make sense even after removing the object "[17]".


Review 3

Summary and Contributions: This paper considers learning the general discrete graphical model. It is an extension of [17] to general basis function.

Strengths: Theoretical results look good and convincing

Weaknesses: It seems like the main difference compared to [17] is to use general basic function in equation (11), instead of focusing on Ising model as in [17]. Other than this, the structure of the paper is similar to [17], e.g. the Local learnability condition corresponds to the restricted strong convexity, e.g. the gradient concentration. The value \hat\gamma in (5) is claimed as the prior information on the parameter (line 271). Based on my understanding it plays a similar role as \lambda in [17]. Is there any particular reason to switch from L1 regularization in [17] to L1 constrained optimization here? Is that possible to provide an order for \hat\gamma? (Theorem 1 in [17], lambda can be chosen as \sqrt(log p/n), a standard order for high dimensional problem). In all, personally, I feel like given [17], the novelty of this work seems a bit weak. The overall model is not changed that much, and the tools used for analysis also looks similar.

Correctness: Seems so

Clarity: The notation is a bit heavy. For example, the underline is used throughout the whole paper. Is that really necessary?

Relation to Prior Work: Yes

Reproducibility: Yes

Additional Feedback:


Review 4

Summary and Contributions: The paper presents a structure and parameter learning framework (and algorithm) for graphical models with discrete domains. The work builds on and generalizes an existing framework called Interaction Screening for Ising Models [Vuffray et al. NeurIPS 2016]

Strengths: One main difficulty with learning undirected graphical models, for the general case, is the identifiability issue (potentials can be parameterized in infinitely many ways) coupled with combinatorial structure identification. My understanding is that the theoretical elements presented in the paper aim to constrain this issue (correct?). For example, the parameterization and structures are heavily regularized. In addition, the aim is to learn the so-called canonical form as alluded in equation (21).

Weaknesses: For the general case, this approach limits the expressibility and utility of the learned model under the canonical form. Although the goal of this paper is to advance our theoretical understanding of learning for (undirected) graphical models (which is fine) my wish (or preference) would be to see some empirical results.

Correctness: They appear correct to me.

Clarity: Paper is well written, but very dense.

Relation to Prior Work: Yes.

Reproducibility: Yes

Additional Feedback: In Algorithm 1, is L an input parameter? If so would it not be difficult to tune? There are many constraints and parameters to identify to make the learning algorithm practical. Also, the parameters in the Theorems are not self contained. ----- update after authors' response ----- I have read through the authors' feedback and reviews. My review remains unchanged.

[Author Response · NeurIPS 2020]

- We would like to start by thanking the authors for their comments and suggestions.

**Reviewer # 1:**

- We also believe that these two elements are the most important points that should be augmented. We will add an
extended discussion about the condition C1 and its relation to the conditional Fisher information matrix and a detailed
discussion with respect to previous work as a conclusion. This will be possible using the 9th page in the camera-ready
version of the paper

**Reviewer # 2:**

- In this paper, we have chosen to focus solely on the theoretical aspects of this algorithm as we believe that a serious
and exhaustive numerical study of the algorithm's performance deserves a dedicated study. For pairwise and binary
variables, [12] shows that RISE beats conditional likelihood and is fast to run (although in [12], the optimization is
performed using a second-order method which is much less efficient than what we propose). We expect GRISE to be a
very efficient algorithm in practice as it contains RISE as a special case.

- The efficient way to verify the identifiability condition is currently in the Appendix D1 and not in the main text. If
space permits, we will put it back in the main text.

- Thank you for the list of typos, we will amend them in the final submission.

**Reviewer # 3:**

- It may look like our paper is a straightforward generalization of [17], but only superficially. This is the result of a
deliberate choice on our part to find a compromise to facilitate the comprehension of this study for readers who are
both familiar and unfamiliar with the field. We understand that this choice may have the unfortunate consequence of
downplaying the innovations presented in this paper. For instance, the local learnability condition looks like it is similar
to the restricted strong convexity condition from [17] but this is not the case. Strong convexity (restricted or not) is
related to the notion of curvature of a function and is therefore linked to the standard Euclidian $\ell_2$-norm. It turns out
that the GRISE loss function is not strongly convex independently of the dimension of the problem (this implies in turns
that the number of samples cannot be $\log p$ according to the standard theory). With the local learnability condition, we
are showing that by looking at the problem using different types of geometry (in our case it is mostly the $\ell_\infty$ geometry)
we can bypass this issue. This approach has been demonstrated using GRISE, but it is in fact an important result in
itself for it is applicable to the analysis of any estimator that minimizes an empirical risk. We also provide a recipe
to efficiently verify this new concept of convexity that is easily applicable to other cost functions. With respect to
other innovations presented in this paper, we would also mention Algorithm 1 which uses a recursive call to GRISE. A
straightforward generalization of [17] will not work for reconstructing graphical models precisely because such a loss
function is not strongly convex independently of the dimension.

- The parameter $\hat{\gamma}$ is a prior on on the strength of the parameters of the problem. This implies that it should be bigger
than the actual $\gamma$ of the system. So unlike the $\lambda$ from [17], $\hat{\gamma}$ does not depend on the size of the system or on the
number of samples. Ideally one would like to take $\hat{\gamma} = \gamma$, but $\gamma$ is an unknown quantity a priori. This implies that in
practice we would chose $\hat{\gamma}$ much larger than the expected $\gamma$. Happily we show that this has only a limited impact on
the sample complexity and the runtime of the algorithm as they both scale polynomially with $\hat{\gamma}$ while it is exponential
in $\gamma$ (which is the best one can hope for). Regardless of the size of the problem, in practice one would not expect $\gamma$
to be bigger than 15 as it means that the intrinsic randomness of the variables is less than 1ppm. There are several
reasons for switching from a penalty (the $\lambda$ form) to a constraint (the $\gamma$ form). The main reason is that the penalty is
designed to enforce sparsity and is adapted to graphical models with bounded degrees. By proposing a constrained
version, we can reconstruct graphical models irrespective of their degree while in [17] dense graphs can require up
to $\mathcal{O}(p^4 \log(p))$ samples. Another reason is that the very nice property of $\lambda$ in [17] being independent of some of the
problem parameters is lost for more general models. In fact $\lambda$ becomes dependant of the parameter $\gamma$ for graphical
models that are not pairwise and binary.

- The graphical models considered here can have a more general form (arbitrary order, arbitrary alphabet, arbitrary
parameterization, arbitrary graph structure) while in [17] it is only applicable to pairwise models with bounded degrees
with binary variables using monomial basis functions. The tools for the analysis may looks superficially similar to [17]
at first glance but are in fact based on a radically new concept that does not use convexity.

**Reviewer # 4:**

- In our study we focus essentially on the theoretical aspect of GM learning (we believe that the current paper is already
very dense). We plan to perform an exhaustive numerical study of this algorithm in a future work.

- In algorithm 1, $L$ is not exactly an input parameter for it is determined by the family of models that is under
consideration and is given by the highest order of the basis functions in the family.

[Meta-Review · NeurIPS 2020]

After discussion, all of the reviewers came to a consensus that the theoretical results in this paper are novel and of significant interest to the NeurIPS community. Please incorporate the comments suggested by the reviewers, paying particular attention to the concerns raised by R3 regarding novelty compared to [17] as well as R1 regarding important discussions and comparisons that should be moved from the supplement to the main paper.